# Mix and Match: An Optimistic Tree-Search Approach for Learning Models from Mixture Distributions

**Matthew Faw**
The University of Texas at Austin
matthewfaw@utexas.edu

**Rajat Sen**
Amazon
rajat.sen@utexas.edu

**Karthikeyan Shanmugam**
IBM Research NY
karthikeyan.shanmugam2@ibm.com

**Constantine Caramanis**
The University of Texas at Austin
constantine@utexas.edu

**Sanjay Shakkottai**
The University of Texas at Austin
sanjay.shakkottai@utexas.edu

## Abstract

We consider a covariate shift problem where one has access to several different training datasets for the same learning problem and a small validation set which possibly differs from all the individual training distributions. The distribution shift is due, in part, to *unobserved* features in the datasets. The objective, then, is to find the best mixture distribution over the training datasets (with only observed features) such that training a learning algorithm using this mixture has the best validation performance. Our proposed algorithm, Mix&Match, combines stochastic gradient descent (SGD) with optimistic tree search and model re-use (evolving partially trained models with samples from different mixture distributions) over the space of mixtures, for this task. We prove a novel high probability bound on the final SGD iterate without relying on a global gradient norm bound, and use it to show the advantages of model re-use. Additionally, we provide simple regret guarantees for our algorithm with respect to recovering the optimal mixture, given a total budget of SGD evaluations. Finally, we validate our algorithm on two real-world datasets.

## 1 Introduction

Suppose a predictive healthcare company has collected data from several regions of the world for a prediction task, and would like to deploy their model in a new region where only preliminary data has been collected. Obtaining more data before product deployment is prohibitively expensive, so they hope to leverage the data they have to deploy their product in this new region. The differences between the distributions from these different regions might arise due to shifts in the observed variables such as weight and height, but might also be caused by shifts in *unobserved variables* not considered or available during data collection, such as prevalence and expression of different conditions, genes, etc.

A natural idea, which allows exploiting the large amount of data from various regions, is to train models on several different mixture distributions over these datasets, and deploy the model that performs best when validated on the small validation data from the new region.

In this paper, we study the problem of correcting for distribution shift using mixture search. Given large datasets from several sources for a common task, and a small validation (enough to validate, but not train a model), we show how to design an algorithm to utilize the large amount of data available for training to train a model performing well on the target distribution.

A notable challenge in our setting is that, in contrast to the typical covariate shift setting, the validation distribution could have shifted due to both observed and *unobserved* variables. As we discuss in Section 4 and Remark 1 common techniques from the covariate shift and domain adaptation literature such as importance weighting [37] and moment matching [13] can fail when these shifts are due in part to shifts in unobserved variables. Hence, our goal is to design a method which is provably robust to such shifts and is also useful in practice.

Perhaps surprisingly, we show that searching over mixtures of training distributions *provably* recovers the optimal model for the validation dataset under mild conditions, *even* when training and validation distribution shift occurs in part due to shifts caused by latent variables (Proposition 1). Further, we show how to efficiently explore this mixture search space by leveraging models trained on near-by mixture distributions. The **main contributions** in this paper are as follows:

**(i) Search based methods for covariate shift:** With latent/unobserved features, we show in Section 4 that traditional methods such as moment matching cannot learn the best mixture distribution (over input datasets) that optimizes performance with respect to a validation set. Instead, we show that searching over input mixture distributions using validation loss results in the recovery of the true model (with respect to the validation, Proposition 1). This motivates our tree search based approach.

**(ii) Mix&Match – Optimistic tree search over models:** We propose Mix&Match – an algorithm that is built on SGD and a variant of optimistic tree-search (closely related to Monte Carlo Tree Search). Given a budget (denoted as $\Lambda$) on the total number of online SGD iterations, Mix&Match adaptively allocates this budget to different population reweightings (mixture distributions over input datasets) through an iterative tree-search procedure (Section 5). Importantly, Mix&Match expends a majority of the SGD iteration budget on reweightings that are "close" to the optimal reweighting mixture by using two important ideas:
*(a) Parsimony in expending iterations:* For a reweighting distribution that we have low confidence of being "good," Mix&Match expends only a small number of SGD iterations to train the model; doing so, however, results in biased and noisy evaluation of this model, due to early stopping in training.
*(b) Re-use of models:* Rather than train a model from scratch, Mix&Match reuses and updates a partially trained model from past reweightings that are "close" to the currently chosen reweighting (effectively re-using SGD iterations from the past).

**(iii) SGD concentrations without global gradient bounds:** The analysis of Mix&Match requires a new concentration bound on the error of the final iterate of SGD. Instead of assuming a uniform bound on the norm of the stochastic gradient over the domain, as is typical in the stochastic optimization literature, we directly exploit properties of the *averaged* loss (strong convexity) and individual loss (smoothness) combined with a bound on the norm of the stochastic gradient *at a single point* to bound the norm of the stochastic gradient at each iterate. Using a single parameter ($\Lambda$, the budget allocated to Mix&Match), we are able to balance the worst-case growth of the norm of the stochastic gradient with the probability of failure of the SGD concentration. This new result (Theorem 5) provides tighter high-probability guarantees on the error of the final SGD iterate in settings where the diameter of the domain is large and/or cannot be controlled.

## 2 Related Work

Transfer learning has assumed an increasingly important role, especially in settings where we are either computationally limited or data-limited but can leverage significant computational and data resources on domains that differ slightly from the target domain [30, 29, 10]. This has become an important paradigm in neural networks and other areas [5, 39, 28, 22]. A related problem is that of covariate shift [36, 40, 13], where the target population distribution may differ from that of the training distribution. Some recent works have considered addressing this problem by reweighting samples from the training dataset so that the distribution better matches the test set, for example by using unlabelled data [20, 13] or variants of importance sampling [37, 38]. The authors in [25] study a related problem of learning from different datasets, but provide minimax bounds in terms of an agnostically chosen test distribution.

Our work is related to, but differs from all the above. As we explain in Section 3, we share the goal of transfer learning: we have access to enough data for training from a family of distributions that are different than the validation distribution (from which we have only enough data to validate). However, to address the effects of latent features, we adopt an optimistic tree search approach – something that, as far as we know, has not been undertaken.

A key component of our tree-search based approach to correcting for distribution shift is the computational budget. We use a single SGD iteration as the currency denomination of our budget, which requires us to minimize the number of SGD steps in total that our algorithm computes, and thus to understand the final-iterate optimization error of SGD in high probability. There are many works deriving error bounds on the final SGD iterate in expectation (e.g. [8, 7, 27]) and in high probability (e.g. [31, 18] and references therein). However, to our knowledge, optimization error bounds on the final iterate of SGD when the stochastic gradient is assumed bounded only at the optimal solution exist only in expectation [27]. We prove a similar result in high probability.

Optimistic tree search makes up the final important ingredient in our algorithm. These ideas have been used in a number of settings [9, 14]. Most relevant to us is a recent extension of these ideas to a setting with biased search [34, 35].

## 3   Problem Setting and Model

**Data model:** Each dataset $\mathcal{D}$ consists of samples of the form $z = (\boldsymbol{x}, y) \in \mathbb{R}^d \times \mathbb{R}$, where $\boldsymbol{x}$ corresponds to the *observed* feature vector, and $y$ is the corresponding label. Traditionally, we would regard dataset $\mathcal{D}$ as governed by a distribution $p(\boldsymbol{x}, y)$. However, we consider the setting where each sample $z$ is a *projection* from some higher dimensional vector $\hat{z} = (\boldsymbol{x}, \boldsymbol{u}, y)$, where $\boldsymbol{u} \in \mathbb{R}^{\hat{d}}$ is the unobserved feature vector. The corresponding distribution function describing the dataset is thus $p(\boldsymbol{x}, \boldsymbol{u}, y)$. This viewpoint allows us to model, for example, predictive healthcare applications where the unobserved features $\boldsymbol{u}$ could represent uncollected, region specific information that is potentially useful in the prediction task (e.g., dietary preferences, workday length, etc.).

We assume access to $K$ **training datasets** $\{\mathcal{D}_i\}_{i=1}^K$ (e.g., data for a predictive healthcare task collected in $K$ different countries) with corresponding p.d.f.'s $\{p_i(\boldsymbol{x}, \boldsymbol{u}, y)\}_{i=1}^K$ through a *sample oracle* to be described shortly. Taking $\triangle := \{\boldsymbol{\alpha} \in \mathbb{R}^K : \boldsymbol{\alpha} \succeq \boldsymbol{0}, \|\boldsymbol{\alpha}\|_1 = 1\}$ as the $(K-1)$-dimensional mixture simplex, for any $\boldsymbol{\alpha} \in \triangle$, we denote the mixture distribution over the training datasets as $p^{(\boldsymbol{\alpha})}(\boldsymbol{x}, \boldsymbol{u}, y) := \sum_{i=1}^K \boldsymbol{\alpha}_i p_i(\boldsymbol{x}, \boldsymbol{u}, y)$. Samples from these datasets may be obtained through a *sample oracle* which, given a mixture $\boldsymbol{\alpha}$, returns an independent sample from the corresponding mixture distribution. In the healthcare example, sampling from $p^{(\boldsymbol{\alpha})}$ would mean first sampling an index $i$ from the multinomial distribution represented by $\boldsymbol{\alpha}$ and then drawing a new medical record from the database of the $i$th country. Additionally, we have access to a *small* (see Remark 5 in the Appendix) validation dataset $\mathcal{D}_{te}$ with corresponding distribution $p^{(te)}(\boldsymbol{x}, \boldsymbol{u}, y)$, for example, from a new country where only limited data has been collected. We are interested in training a predictive model for the validation distribution, but we *do not* have oracle sampling access to this distribution – if we did, we could simply train a model through SGD directly on this dataset. Instead, we only assume *oracle access to evaluating the validation loss of a constrained set of models* (we define our loss model and the constrained set shortly).

**Loss function model:** We denote the loss for a particular sample $z$ and model $\boldsymbol{w} \in \mathcal{W} := \mathbb{R}^m$ as $f(\boldsymbol{w}; z)$. For any mixture distribution $\boldsymbol{\alpha} \in \triangle$, we denote $F^{(\boldsymbol{\alpha})}(\boldsymbol{w}) := \mathbb{E}_{z \sim p^{(\boldsymbol{\alpha})}}[f(\boldsymbol{w}; z)]$ as the *averaged* loss function over distribution $p^{(\boldsymbol{\alpha})}$. Note that when $\boldsymbol{\alpha}$ is clear from context, we write $F(\boldsymbol{w})$. Similarly, we denote $F^{(te)}(\boldsymbol{w}) := \mathbb{E}_{z \sim p^{(te)}}[f(\boldsymbol{w}; z)]$ as the averaged validation loss. We place the following assumptions on our loss function, similar to [27] (refer to Appendix B for these standard definitions):

**Assumption 1.** *For each loss function $f(\cdot; z)$ corresponding to a **sample** $z \in \mathcal{Z}$, we assume that $f(\cdot; z)$ is $\beta$-**smooth** and **convex**. Additionally, we assume that, for each $\boldsymbol{\alpha} \in \triangle$, the **averaged** loss function $F^{(\boldsymbol{\alpha})}(\cdot)$ is $\mu$-**strongly convex**.*

Notice that Assumption 1 requires only the *averaged* loss function $F^{(\boldsymbol{\alpha})}(\cdot)$ – *not* each individual loss function $f(\cdot; z)$ – to be strongly convex. Additionally, we assume the stochastic gradient satisfies:

**Assumption 2** (A *weaker* gradient norm bound). *For all $\boldsymbol{\alpha} \in \triangle$, there exists constants $\mathcal{G}_*(\boldsymbol{\alpha}) \in \mathbb{R}_+$ such that $\|\nabla f(\boldsymbol{w}^*(\boldsymbol{\alpha}); z)\|_2^2 \leq \mathcal{G}_*(\boldsymbol{\alpha})$. When $\boldsymbol{\alpha}$ is clear from context, we write $\mathcal{G}_*$.*

We note that Assumption 2 is *weaker* than the typical universal bound on the norm of the stochastic gradient assumed in, for example, [18], and is taken from [27].

## 4 Problem Formulation

Given $K$ training datasets $\{\mathcal{D}_i\}_{i=1}^K$ (e.g., healthcare data from $K$ countries) and a small (see Remark 5), labeled validation dataset $\mathcal{D}_{te}$ (e.g., preliminary data collected in a new country), we wish to find a model $\hat{\boldsymbol{w}}$ such that the loss averaged over the validation *distribution*, $p^{(te)}$, is as small as possible, using a computational budget to be described shortly. Under the notation introduced in Section 3, we wish to approximately solve the optimization problem:

$$\min_{\boldsymbol{w} \in \mathcal{W}} F^{(te)}(\boldsymbol{w}), \tag{1}$$

subject to a **computational budget of $\Lambda$ SGD iterations** and constrained such that $F^{(te)}(\cdot)$ can be evaluated *only* at models $\boldsymbol{w}$ obtained using at least 1 SGD iteration. A computational budget is often used in online optimization as a model for constraints on the number of i.i.d. samples available to the algorithm (see for example the introduction to Chapter 6 in [8]).

Note that one *could* run SGD directly on the validation dataset, $\mathcal{D}_{te}$, in order to minimize the expected loss on this population, as long as the number of SGD steps is *linear* in the size of $\mathcal{D}_{te}$ [17]. When the number of validation samples is *small relative to the computational budget* $\Lambda$, such as in the predictive healthcare example where little data from the new target country is available, the resulting error guarantees of such a procedure will be correspondingly weak. Thus, we hope to leverage both training data *and* validation data in order to solve Equation 1.

Though we cannot train a model using $\mathcal{D}_{te}$, we will assume $\mathcal{D}_{te}$ is sufficiently large to obtain an accurate estimate of the validation loss. We model evaluations of validation loss through oracle access to $F^{(te)}(\cdot)$, which may be queried **only on models trained by running at least one SGD iteration on some mixture distribution over the training datasets.**

Let $\boldsymbol{w}^*(\boldsymbol{\alpha}) := \arg\min_{\boldsymbol{w} \in \mathcal{W}} F^{(\boldsymbol{\alpha})}(\boldsymbol{w})$ be the optimal model for training mixture distribution $p^{(\boldsymbol{\alpha})}$. Similarly, let us denote $\hat{\boldsymbol{w}}(\boldsymbol{\alpha})$ as the model obtained after running $1 \leq T \leq \Lambda$ steps of online SGD on $p^{(\boldsymbol{\alpha})}$. Then we can *minimize validation loss* $F^{(te)}(\cdot)$ by **(i)** *iteratively selecting mixtures* $\boldsymbol{\alpha} \in \triangle$, **(ii)** using a *portion* of the SGD budget to *solve for* $\hat{\boldsymbol{w}}(\boldsymbol{\alpha})$, and **(iii)** *evaluating the quality of the selected mixture* by obtaining the validation loss $F^{(te)}(\hat{\boldsymbol{w}}(\boldsymbol{\alpha}))$ (through oracle access, as discussed earlier). That is, using $\Lambda$ total SGD iterations, we can find a *mixture distribution* $\boldsymbol{\alpha}(\Lambda)$ and *model* $\hat{\boldsymbol{w}}(\boldsymbol{\alpha}(\Lambda))$ so that $F^{(te)}(\hat{\boldsymbol{w}}(\Lambda))$ is as close as possible to

$$\min_{\boldsymbol{\alpha} \in \triangle} G(\boldsymbol{\alpha}) = \min_{\boldsymbol{\alpha} \in \triangle} F^{(te)}(\boldsymbol{w}^*(\boldsymbol{\alpha})), \tag{2}$$

where $G(\boldsymbol{\alpha}) := F^{(te)}(\boldsymbol{w}^*(\boldsymbol{\alpha}))$ is the test loss evaluated at the *optimal model for* $p^{(\boldsymbol{\alpha})}$.

Under rather general conditions, we can show that solving Equation 1 and Equation 2 are actually equivalent:

**Proposition 1.** *Suppose that $p^{(\boldsymbol{\alpha}^*)}(\boldsymbol{x}, \boldsymbol{u}) = p^{(te)}(\boldsymbol{x}, \boldsymbol{u})$ for some mixture distribution $\boldsymbol{\alpha}^* \in \triangle$, and additionally that $p_i(y|\boldsymbol{x}, \boldsymbol{u}) = p^{(te)}(y|\boldsymbol{x}, \boldsymbol{u})$ for every $i \in [K]$. Then validation loss can be written in terms of mixtures of training loss:*

$$F^{(te)}(\boldsymbol{w}) = F^{(\boldsymbol{\alpha}^*)}(\boldsymbol{w}) \tag{3}$$

*for each $\boldsymbol{w} \in \mathcal{W}$. As a consequence, finding $\boldsymbol{w}^*$ which solves Equation 1 is **equivalent** to finding the mixture $\boldsymbol{\alpha}^*$ and corresponding model $\boldsymbol{w}^*(\boldsymbol{\alpha}^*)$ which solves Equation 2, since $F^{(te)}(\boldsymbol{w}^*) = F^{(\boldsymbol{\alpha}^*)}(\boldsymbol{w}^*) = F^{(\boldsymbol{\alpha}^*)}(\boldsymbol{w}^*(\boldsymbol{\alpha}^*))$.*

Proposition 1 follows immediately by noting that, $p^{(te)}(\boldsymbol{x}, \boldsymbol{u}, y) = p^{(\boldsymbol{\alpha}^*)}(\boldsymbol{x}, \boldsymbol{u}, y)$, and thus $p^{(te)}(\boldsymbol{x}, y) = p^{(\boldsymbol{\alpha}^*)}(\boldsymbol{x}, y)$, and by using the definition of $F^{(te)}$.

We take as our objective to minimize simple regret with respect to the optimal model $\boldsymbol{w}^*(\boldsymbol{\alpha}^*)$:

$$R(\Lambda) := G(\boldsymbol{\alpha}(\Lambda)) - \min_{\boldsymbol{\alpha} \in \triangle} G(\boldsymbol{\alpha}). \tag{4}$$

That is, we measure the performance of our algorithm by the difference in validation loss between *the best model corresponding to our final selected mixture, $\boldsymbol{w}^*(\boldsymbol{\alpha}(\Lambda))$* and the best model for the validation loss, $\boldsymbol{w}^*(\boldsymbol{\alpha}^*)$.

Note that the conditions placed on the distributions $p_i$ in Proposition 1 are quite general. Indeed, they generalize the settings considered in multi-source domain adaptation [24] (where the assumption is that $p^{(te)}(\boldsymbol{x}, y) = p^{(\boldsymbol{\alpha}^*)}(\boldsymbol{x}, y)$ for some $\boldsymbol{\alpha}^* \in \triangle$) as well as the covariate shift assumption from, e.g., [19, 36] (where the assumption is that $p^{(te)}(y|\boldsymbol{x}) = p_i(y|\boldsymbol{x})$). Thus, we are able to address a *novel setting* in which distribution shift can be attributed to both observed *and unobserved* features.

Further, one should not view a specific choice of $K$ data sources as an inherent restriction. One could split this dataset into $K' > K$ sources (even if $K = 1$) by running an unsupervised clustering algorithm to create new data sources grouped by some notion of feature similarity.

**Remark 1** (**Difficulties with moment matching and domain invariant representations**). *Note that we cannot learn $\boldsymbol{\alpha}^*$ simply by matching the mixture distribution over the training sets to that of the validation set (both with only the observed features and labels). This is because $p_k(x, u)$ decomposes as $p_k(x, u) = p_k(x)p_k(u|x)$, where $p_k(u|x)$ is unknown and potentially differs across datasets. Thus, in a setting with unobservable features, approaches that try to directly learn the mixture weights by comparing with the validation set (e.g., using an MMD distance or moment matching) learns the wrong mixture weights. Further, our scenario also admits cases where the observed $p(y|x)$ (label distribution conditioned on observed variables) can shift which is non-trivial. In fact, when observed conditional distribution of labels differ between training and validation, strong lower bounds exist on many variants of another popular method called domain invariant representation (see Corollary 4.1 in [41]).*

## 5   Theoretical Foundations for Algorithm

We now present Mix&Match (Algorithm 1), our proposed algorithm for minimizing $G(\boldsymbol{\alpha}) = F^{(te)}(\boldsymbol{w}^*(\boldsymbol{\alpha}))$ over the mixture simplex $\triangle$ using $\Lambda$ total SGD iterations. To solve this minimization problem, our algorithm must search over the mixture simplex, and for each $\boldsymbol{\alpha} \in \triangle$ selected by the algorithm, *approximately* evaluate $G(\boldsymbol{\alpha})$ by obtaining an approximate minimizer $\hat{\boldsymbol{w}}(\boldsymbol{\alpha})$ of $F^{(\boldsymbol{\alpha})}(\cdot)$ and evaluating $\widehat{G}(\boldsymbol{\alpha}) = F^{(\boldsymbol{\alpha})}(\hat{\boldsymbol{w}}(\boldsymbol{\alpha}))$. Two main ideas underlie our algorithm: **parsimony in expending SGD iterations** – using a small number of iterations for mixture distributions that we have a low confidence are "good" – and **model reuse** – using models trained on nearby mixtures as a starting point for training a model on a new mixture distribution. We now outline why and how the algorithm utilizes these two ideas.

**Warming up: model search with optimal mixture.** By Proposition 1, $G(\boldsymbol{\alpha}) = F^{(\boldsymbol{\alpha}^*)}(\boldsymbol{w}^*(\boldsymbol{\alpha}))$ for all $\boldsymbol{\alpha} \in \triangle$. Therefore, if we were given $\boldsymbol{\alpha}^*$ a priori, then we could run stochastic gradient descent to minimize the loss over this mixture distribution on the training datasets, $F^{(\boldsymbol{\alpha}^*)}(\cdot)$, in order to find an $\varepsilon$-approximate solution to $\boldsymbol{w}^*(\boldsymbol{\alpha}^*)$, the desired optimal model for the validation distribution. In our experiments (Section 6 and Appendix H), we will refer to this algorithm as the Genie. *Our algorithm, thus, will be tasked to find a mixture close to $\boldsymbol{\alpha}^*$.*

**Close mixtures imply close optimal models.** Now, suppose that instead of being given $\boldsymbol{\alpha}^*$, we were given some other $\hat{\boldsymbol{\alpha}} \in \triangle$ which is close to $\boldsymbol{\alpha}^*$ in $\ell_1$ distance. The following theorem tells us that optimal models for these two mixtures will also be close:

**Theorem 1.** *Consider a loss function $f(\boldsymbol{w}; z)$ which satisfies Assumption 1 and Assumption 2, and the convex body $\mathcal{X} = \mathrm{Conv}\{\boldsymbol{w}^*(\boldsymbol{\alpha}) \in \mathcal{W}|\boldsymbol{\alpha} \in \triangle\}$. Then for any $\boldsymbol{\alpha}_1, \boldsymbol{\alpha}_2 \in \triangle$, $\|\boldsymbol{w}^*(\boldsymbol{\alpha}_1) - \boldsymbol{w}^*(\boldsymbol{\alpha}_2)\|_2 \leq \frac{2\sigma}{\mu}\|\boldsymbol{\alpha}_1 - \boldsymbol{\alpha}_2\|_1$, where $\sigma^2 = \sup_{\substack{\boldsymbol{w}, \boldsymbol{w}' \in \mathcal{X} \\ \boldsymbol{\alpha} \in \triangle}} \beta^2 \|\boldsymbol{w} - \boldsymbol{w}'\|_2^2 + \mathcal{G}_*(\boldsymbol{\alpha})$.*

The above theorem is essentially a generalization of Theorem 3.9 in [17] to the case when only $\mathbb{E}[f]$, not $f$, is strongly convex. *As a consequence of Theorem 1, our algorithm needs only to find a mixture $\hat{\boldsymbol{\alpha}}$ sufficiently close to $\boldsymbol{\alpha}^*$.*

**Smoothness of $G(\cdot)$ and existence of "good" simplex partitioning implies applicability of optimistic tree search algorithms.** This notion of smoothness of $G(\boldsymbol{\alpha})$ immediately implies that we can use the optimistic tree search framework similar to [9, 14] in order to minimize $G(\boldsymbol{\alpha})$ by performing a tree search procedure over hierarchical partitions of the mixture simplex $\triangle$ – indeed, in this literature, such smoothness conditions are directly assumed. Additionally, the existence of a hierarchical partitioning such that the diameter of each partition cell decays exponentially with tree height is also assumed. In our work, however, we prove that the smoothness condition on $G(\cdot)$ holds, and by using the simplex bisection strategy described in [21], the cell diameter decay condition also holds, making tree search a natural algorithm to choose:

**Corollary 1** (of Theorem 1). *There exists a hierarchical partitioning $\mathcal{P}$ of the simplex of mixture weights $\triangle$ (namely, the simplex bisection strategy described in [21]), such that, for any cell $(h,i) \in \mathcal{P}$, and any $\boldsymbol{\alpha}_1, \boldsymbol{\alpha}_2 \in (h,i)$,*

$$\|\boldsymbol{\alpha}_1 - \boldsymbol{\alpha}_2\|_1 \leq \sqrt{2K} \left( \frac{\sqrt{3}}{2} \right)^{\frac{h}{K-1} - 1}$$

*where $K - 1 = \dim(\triangle)$. Combined with Theorem 1, this implies that $\|\boldsymbol{w}^*(\boldsymbol{\alpha}_1) - \boldsymbol{w}^*(\boldsymbol{\alpha}_2)\|_2^2 \leq \nu_1 \rho^h$, and $|G(\boldsymbol{\alpha}_1) - G(\boldsymbol{\alpha}_2)| \leq \nu_2 \rho_2^h$, where $\nu_1 = \left( \frac{4\sigma\sqrt{2K}}{\sqrt{3}\mu} \right)^2$, $\rho = \left( \frac{\sqrt{3}}{2} \right)^{\frac{2}{K-1}}$, $\nu_2 = \sqrt{\nu_1} \sup_{\boldsymbol{\alpha} \in \triangle} \|\boldsymbol{w}^*(\boldsymbol{\alpha}) - \boldsymbol{w}^*(\boldsymbol{\alpha}^*)\|_2$, and $\rho_2 = \sqrt{\rho}$.*

*Thus, it is natural to design our algorithm in the tree search framework.*

**Tree search framework.** Mix&Match proceeds by constructing a binary partitioning tree $\mathcal{T}$ over the space of mixtures $\triangle$. Each node $(h,i) \in \mathcal{T}$ is indexed by the height (i.e. distance from the root node) $h$ and the node's index $i \in [2^h]$ in the layer of nodes at height $h$. The set of nodes $V_h = \{(h,i) : i \in [2^h]\}$ at height $h$ are associated with a partition $\mathcal{P}_h = \{\mathcal{P}_{h,i} : i \in [2^h]\}$ of the mixture simplex $\triangle$ into $2^h$ disjoint partition cells whose union is $\triangle$. The root node $(0,1)$ is associated with the entire simplex $\triangle$, and two children of node $(h,i)$, $\{(h+1, 2i-1), (h+1, 2i)\}$ correspond to the two partition cells of the parent node's partition. The resulting hierarchical partitioning will be denoted $\mathcal{P} = \cup_h \mathcal{P}_h$, and can be implemented using the simplex bisection strategy of [21]. *Combined with the smoothness results on our objective function, $\mathcal{T}$ gives a natural structure to search for $\boldsymbol{\alpha}^*$.*

**Multi-fidelity evaluations of $G(\cdot)$ – associating $\mathcal{T}$ with mixtures and models.** We note that, in our setting, $G(\boldsymbol{\alpha}) = F^{(te)}(\boldsymbol{w}^*(\boldsymbol{\alpha}))$ cannot be directly evaluated, since we cannot obtain $\boldsymbol{w}^*(\boldsymbol{\alpha})$ explicitly, but only an approximate minimizer $\hat{\boldsymbol{w}}(\boldsymbol{\alpha})$. Thus, we take inspiration from recent works in *multi-fidelity* tree-search [34, 35]. Specifically, using a height-dependent SGD budget function $\lambda(h)$, the algorithm takes $\lambda(h)$ SGD steps using some selected mixture $\boldsymbol{\alpha}_{h,i} \in \mathcal{P}_{h,i}$ to obtain an approximate minimizer $\hat{\boldsymbol{w}}(\boldsymbol{\alpha}_{h,i})$ and evaluates the validation loss $F^{(te)}(\hat{\boldsymbol{w}}(\boldsymbol{\alpha}_{h,i}))$ to obtain an estimate for $G(\boldsymbol{\alpha}_{h,i})$. *$\lambda(\cdot)$ is designed so that estimates of $G(\cdot)$ are "crude" early during the tree-search procedure and more refined deeper in the search tree.*

**Warm starting with the parent model.** When our algorithm, Mix&Match selects node $(h,i)$, it creates child nodes $\{(h+1, 2i-1), (h+1, 2i)\}$, and runs SGD on the associated mixtures $\boldsymbol{\alpha}_{h+1,2i-1}$ and $\boldsymbol{\alpha}_{h+1,2i}$, starting each SGD run with initial model $\hat{\boldsymbol{w}}(\boldsymbol{\alpha}_{h,i})$, *the final iterate of the parent node's SGD run*. Since $\boldsymbol{\alpha}_{h,i}$ and $\boldsymbol{\alpha}_{h+1,j}$ ($j \in \{2i-1, 2i\}$) are exponentially close as a function of $h$ (as a consequence of our simplex partitioning strategy), so too are $\boldsymbol{w}^*(\boldsymbol{\alpha}_{h,i})$ and $\boldsymbol{w}^*(\boldsymbol{\alpha}_{h+1,j})$ (since close mixtures implies close models). Thus, as long as the parent's final iterate is exponentially close to $\boldsymbol{w}^*(\boldsymbol{\alpha}_{h,i})$, then the initial iterate for the SGD runs associated to the child nodes will also be exponentially close to their associated solution, $\boldsymbol{w}^*(\boldsymbol{\alpha}_{h+1,j})$. *Therefore, a good initial condition of weights for a child node's model is that resulting from the final iterate of the parent's model.*

**Constant SGD steps suffice for exponential error improvement.** In a noiseless setting (e.g., the setting of Theorem 3.12 in [8]), optimization error scales linearly in the squared distance between the initial model and the optimal model, and thus, in this setting, we could simply take a constant number of gradient descent steps to obtain a model with error decaying exponentially in $h + 1$. However, in SGD, optimization error depends not only on the initial distance to the optimal model, but also on the noise of the stochastic gradient. While some SGD results give guarantees *in expectation* which scale with the distance between the initial and optimal iterate (e.g., Theorem 4.7 in [7]), existing high-probability results (i) rely on a *global* bound on the norm of the stochastic gradient and

(ii) scale with the diameter of the search space (e.g., [18]). We are able to establish the following *high-probability* SGD result for the *final* SGD iterate which captures the error scaling with respect to the distance of the initial iterate from the optimal model $d_0(\boldsymbol{\alpha}) = \|\boldsymbol{w}_0(\boldsymbol{\alpha}) - \boldsymbol{w}^*(\boldsymbol{\alpha})\|_2$, the norm of the stochastic gradient **only at the optimal solution** $\mathcal{G}_*$, and the global diameter bound $D$:

**Theorem 2** (Informal statement of Theorem 5). *If we run SGD for $t + 1$ steps to minimize training loss over mixture distribution $\boldsymbol{\alpha}$ starting from a fixed vector $\boldsymbol{w}_0 \in \mathcal{W}$ and using decreasing step size $\eta_t = \frac{2}{\mu(t+E)}$, where $E = 4096\kappa^2 \log \Lambda^8$ and $\kappa = \frac{\beta}{\mu}$ is the condition number, then with probability at least $1 - \frac{t+1}{\Lambda^8}$, for any $k \in \mathbb{Z}_{\geq 0}$, the error of the last iterate satisfies*

$$\|\boldsymbol{w}_{t+1} - \boldsymbol{w}^*\|_2^2 \leq \underbrace{\frac{\Gamma(d_0^2, \mathcal{G}_*)}{t + E + 1}}_{\substack{\text{bound on } \mathbb{E}[d_{t+1}^2] \\ \text{from [27]}}} + \underbrace{\frac{\tilde{C}(D^2, D\sqrt{\mathcal{G}_*})}{\mu \Lambda^7}}_{\substack{\text{Global diameter bound} \\ \text{controlled by } \Lambda}} + \underbrace{\frac{4\sqrt{2\log(\Lambda^8)}\sqrt{\hat{C}(k; d_0, \mathcal{G}_*)}}{\mu(t + E + 1)^{\alpha_{k+1}}}}_{\text{term to control martingale deviations}}$$

*where $\Gamma(d_0^2, \mathcal{G}_*) = \max\{Ed_0^2, 8\mathcal{G}_*/\mu^2\}$, $\alpha_{k+1} = \sum_{i=1}^{k+1} \frac{1}{2^i} \in [\frac{1}{2}, 1)$ and $\hat{C}(k; d_0, \mathcal{G}_*)$ scales polynomially in $\mathcal{G}_*$ and $d_0$, where the polynomial dependence is* independent *of $k$.*

The crucial insight that allows this bound without the global stochastic gradient bound is that, under our $\beta$-smoothness assumption and Assumption 2, however, we can show that, until we hit the noise floor of $\mathcal{G}_*(\boldsymbol{\alpha})$ (the bound on the norm of the gradient *only at the optimal model* $\boldsymbol{w}^*(\boldsymbol{\alpha})$), the noise of the stochastic gradient *also decays exponentially with tree height* (see e.g. Lemma 3 in the Appendix for a proof). *As a consequence, until we hit this noise floor, we may take a constant number of SGD steps to exponentially improve the optimization error as we descend our search tree (refer to Corollary 3 in the Appendix for details). In fact, all of our experiments (Section 6 and Appendix H) use a height-independent budget function $\lambda$.*

**Growing the search tree.** Now we present our final bound that characterizes the performance of Mix&Match as Theorem 3. In the deterministic black-box optimization literature [26, 34], the quantity of interest is generally *simple regret*, $R(\Lambda)$, as defined in Equation 4. In this line of work, the simple regret scales as a function of *near-optimality dimension*, which is defined as follows [14]:

**Definition 1.** *The near-optimality dimension of $G(\cdot)$ with respect to parameters $(\nu_1, \rho)$ is given by:*

$$d(\nu_1, \rho) = \inf\left\{d' \in \mathbb{R}^+ : \exists\, C(\nu_1, \rho), \text{ s.t. } \forall h \geq 0, \mathcal{N}_h(3\tilde{\nu}\rho_2^h) \leq C(\nu_1, \rho)\rho_2^{-d'h}\right\},$$

*where $\mathcal{N}_h(\epsilon)$ is the number of cells $(h, i)$ such that $\inf_{\boldsymbol{\alpha} \in (h,i)} G(\boldsymbol{\alpha}) \leq G(\boldsymbol{\alpha}^*) + \varepsilon$, $\rho_2 = \sqrt{\rho}$, $\tilde{\nu} = \sqrt{\nu_1}(L + \frac{\beta\sqrt{\nu_1}}{6})$, and $L = \beta \sup_{\boldsymbol{\alpha} \in \triangle} \|\boldsymbol{w}^*(\boldsymbol{\alpha}) - \boldsymbol{w}^*(\boldsymbol{\alpha}^*)\|_2$.*

The near-optimality dimension intuitively states that there are *not too many* cells which contain a point whose function values are *close* to optimal *at any tree height*. The lower the near-optimality dimension, the easier is the black-box optimization problem [14]. Theorem 3 provides a similar simple regret bound on $R(\Lambda) = G(\boldsymbol{\alpha}(\Lambda)) - G(\boldsymbol{\alpha}^*)$, where $\boldsymbol{\alpha}(\Lambda)$ is the mixture weight vector returned by the algorithm given a total SGD steps budget of $\Lambda$ and $\boldsymbol{\alpha}^*$ is the optimal mixture. The proof of Theorem 3 is in Appendix E.

**Theorem 3.** *Let $h'$ be the smallest number $h$ such that $\sum_{l=0}^{h} 2C(\nu_1, \rho)\lambda(l)\rho_2^{-d(\nu_1,\rho)l} > \Lambda - 2\lambda(h+1)$. Then, with probability at least $1 - \frac{1}{\Lambda^3}$, the tree in Algorithm 1 grows to a height of at least $h(\Lambda) = h' + 1$ and returns a mixture weight $\boldsymbol{\alpha}(\Lambda)$ such that*

$$R(\Lambda) \leq 4\hat{\nu}\rho_2^{h(\Lambda)-1} \tag{5}$$

*where $\hat{\nu} = \sqrt{\nu_1}\left(L + \frac{\beta}{8}\sqrt{\nu_1}\right)$, and $L = \beta \sup_{\boldsymbol{\alpha} \in \triangle} \|\boldsymbol{w}^*(\boldsymbol{\alpha}) - \boldsymbol{w}^*(\boldsymbol{\alpha}^*)\|_2$.*

Theorem 3 shows that, given a total budget of $\Lambda$ SGD steps, Mix&Match recovers a mixture $\boldsymbol{\alpha}(\Lambda)$ with test error at most $4\hat{\nu}\rho_2^{h(\Lambda)-1}$ away from the optimal test error if we perform optimization using that mixture. The parameter $h(\Lambda)$ depends on the number of steps needed for a node expansion at different heights and crucially makes use of the fact that the starting iterate for each new node can be borrowed from the parent's last iterate. The tree search also progressively allocates more samples

**Algorithm 1** Mix&Match (Simplified – details in Algorithm 2): Tree-Search over the mixtures of training datasets

---

**Input:** Real $\rho, \rho_2 \in (0,1)$, $\nu_1 > 0$ and hierarchical partition $\mathcal{P}$ of $\triangle$ as specified in Corollary 1, $\nu = \sqrt{\nu_1} \left( \frac{\beta \sqrt{\nu_1}}{2} + L \right)$ total SGD budget for *entire tree search procedure* $\Lambda > 0$, initial model $\boldsymbol{w}_0 \in \mathcal{W}$.

1: Initialize search tree $\mathcal{T}_0 = \{(0,1)\}$ with initial model $\widehat{\boldsymbol{w}}(\boldsymbol{\alpha}_{0,1})$ trained using SGD (from Theorem 2) on training mixture distribution $\boldsymbol{\alpha}_{0,1} \in \mathcal{P}_{0,1}$ to optimization error $2\nu_1 \rho^0$. Denote $\lambda(0)$ as the number of SGD steps taken.
2: Cost (Number of SGD steps used): $C = \lambda(0)$
3: **while** $C \leq \Lambda$ **do**
4:    Select the leaf $(h,j) \in \text{Leaves}(\mathcal{T}_t)$ with minimum $b_{h,j} := F^{(te)}(\widehat{\boldsymbol{w}}(\boldsymbol{\alpha}_{h,j})) - 2\nu \rho_2^h$.
5:    Add to $\mathcal{T}_t$ the 2 children of $(h,j)$ (as determined by $\mathcal{P}$) by running SGD on two mixtures from the two child partitions of $\mathcal{P}_{h,i}$ to obtain optimization error $2\nu_1 \rho^{h+1}$.
6:    $C = C + 2\lambda(h+1)$.
7: **end while**
8: Let $h(\Lambda)$ be the height of $\mathcal{T}_t$
9: Let $i^* := \arg\min_i F^{(te)}(\widehat{\boldsymbol{w}}(\boldsymbol{\alpha}_{h(\Lambda),i}))$.
10: Return $\boldsymbol{\alpha}_{h(\Lambda),i^*}$ and $\widehat{\boldsymbol{w}}(\boldsymbol{\alpha}_{h(\Lambda),i^*})$.

---

to deeper nodes, as we get closer to the optimum. Similar simple regret scalings have been recently shown in the context of deterministic multi-fidelity black-box optimization [34]. Note that Theorem 3 roughly corresponds to a regret scaling on the order of $\tilde{O}\left(\frac{1}{\Lambda^c}\right)$ for some constant $c$ (dependent on $d(\nu_2, \rho_2)$). Thus, when $|\mathcal{D}_{te}|$ is much smaller than the total computational budget $\Lambda$, our algorithm gives a significant improvement over training only on the validation dataset. In our experiments in Section 6 and Appendix H, we observe that our algorithm indeed outperforms the algorithm which trains only on the validation dataset for several different real-world datasets.

# 6 Empirical Results

**Algorithms compared:** We compare the following algorithms: (a) `Uniform` trains on samples from each data source uniformly, (b) `Genie` samples from training data sources according to $\boldsymbol{\alpha}^*$ in those cases when $\boldsymbol{\alpha}^*$ is known explicitly (*this can be viewed as the best-case comparison for our algorithm, since it already knows* $\boldsymbol{\alpha}^*$), (c) `Validation` trains only on samples from the validation dataset, (d) `Mix&MatchCH+0.1Step` runs Mix&Match by partitioning the $\boldsymbol{\alpha}$ simplex using a random coordinate halving strategy for half of the budget and using the remaining half of the budget to train the model on the mixture distribution selected by Mix&Match (with step size multiplied by $0.1$), (e) `OnlyX` trains on samples *only* from data source X, (f) `IW-Uniform` computes importance weights by training a logistic regression model then runs importance-weighted `Uniform`, (g) `IW-ERM` similarly computes importance weights and then runs importance-weighted empirical risk minimization, and (h) `MMD` constructs a representative training set by computing the MMD metric using validation data. We describe results with other Mix&Match algorithm variants in Appendix H.

**Models and metrics:** We use fully connected neural networks with ReLU activations for all our experiments, training with cross-entropy loss on the categorical labels. We use the test AUROC as the metric for comparison between the above mentioned algorithms. For multiclass problems, we use multiclass AUROC metric described in [15]. The reason for using AUROC is due to the label imbalances due to covariate shifts between the training sources and our test and validation sets. Since all algorithms we compare against use SGD, we use the same hyperparameters (SGD step size, neural network architecture, etc.) across all algorithms for each dataset/plot (however, they are different across plots/datasets). Experiment code is availale here.

**Plot details:** In the experiments displayed below, we plot the performance of each algorithm at *intermediate measurement points*, where each displayed data point is the result of averaging over 10 experiments, with error bars of 1 standard deviation. That is, for each algorithm, we specify a total SGD budget (60k for the Allstate experiment, and 200k for the MNIST experiment), and report the test AUC measured at intermediate intervals throughout each experiment's duration.

**Allstate Purchase Prediction Challenge:** The Allstate Purchase Prediction Challenge Kaggle dataset [1] has entries from customers across different states in the US. The goal is to predict

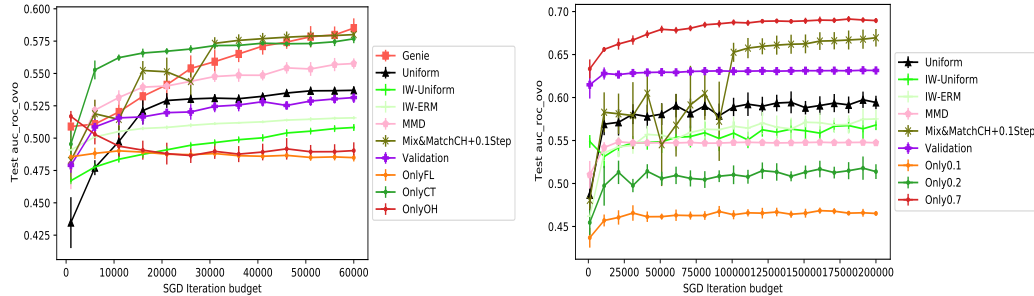

(a) Test AUROC for predicting insurance plan for a mixture of FL, CT, and OH data

(b) Test AUROC for colored MNIST digit $< 5$ for test environment color flip probability $0.6$

Figure 1: Experimental results for Allstate experiment (left) and MNIST experiment (right)

what option a customer would choose for an insurance plan in a specific category (Category G with $4$ options ). The dataset features include (a) demographics and details regarding vehicle ownership of a customer and (b) timestamped information about insurance plan selection across seven categories (A-G) used by customers to obtain price quotes. There are multiple timestamped category selections and corresponding price quotes for a customer. We collapse the selections and the price quote to a single set of entries using summary statistics of the time stamped features.

In this experiment, we split the Kaggle dataset into $K = 3$ training datasets correspond to customer data from three states: Florida (FL), Connecticut (CT), and Ohio (OH). The validation and test datasets also consist of customers from these states, but the proportion of customers from various states is fixed. Details about the test and validation set formation is in the Appendix. In this case, $\boldsymbol{\alpha}^*$ is explicitly known for the `Genie` algorithm.

As shown in Figure 1a, with respect to the AUROC metric, `Mix&MatchCH` is competitive with the `Genie` algorithm and has superior performance to all other baselines. The `Validation` algorithm has performance inferior to the uniform sampling scheme. Therefore, we are operating in a regime in which training on the validation set alone is not sufficient for good performance. Note that, since the validation mixture is mostly ($\sim 93\%$) CT data (see Table 2 in the Appendix), it is reasonable that `OnlyCT` outperforms `Genie` during earlier iterations.

**Colored MNIST:** We evaluate our algorithms on an MNIST digit binary classification problem to predict whether a given digit is smaller than or larger than $5$. Similarly to the colored MNIST experiment from [2], each label is flipped with probability $0.25$ (so a classifier ignoring color should have accuracy $0.75$), the images are colored one of two colors according to the label of the image, and then the color is flipped with probability $e_i \in \{0.1, 0.2, 0.7\}$ for data source $i \in [3]$. Thus, each training environment has spurious color correlations with the label, with the majority of training labels positively correlated with the labels, while the validation environment has negative correlation with the labels, as the color in this environment is flipped with probability $0.6$.

In Figure 1b, we observe that, since the training environments have labels both positively *and* negatively correlated with color, Mix&Match trains a model which outperforms the other baselines, and has performance competitive with `Only0.7`, which is trained on a classifier negatively correlated with color, even though the majority of training examples have the opposite color correlation. (Note that we do not compare against the `Genie` in this experiment, since $\boldsymbol{\alpha}^*$ is not explicitly given).

We note that, in Figure 1b, Mix&Match has a seemingly sudden jump in AUC after half of the SGD budget has been used. A closer examination of Figure 1a shows a similar phenomenon, again at the half SGD budget mark ($30k$). Recall that the implementation of Mix&Match in our experiments, `Mix&MatchCH+0.1Step`, devotes *half* of its budget to tree-search, and *half* of its budget to optimize the *best* model output by tree-search. The measurement points for Mix&Match during the first half of the SGD budget correspond to the AUC of the model *currently being trained by* Mix&Match, which might *not* be the best model selected at the end of the tree-search. Thus, we should expect that the AUC of the model being trained by `Mix&MatchCH+0.1Step` to be *higher* in the second half of the SGD budget than in the first half. We refer to Appendix H for further details and experiments.

## Broader Impact

This work addresses the problem of using abundant training data from several domains, along with a limited amount of data from a new, related domain, to train effective models for this new domain. As we discussed in the introduction, a natural motivation for this setup is that a company wants to make predictions on a new population, but most data available at training time is not drawn from this target distribution. Our focus in this work is primarily theoretical – we design an algorithm that can provably succeed at training a good model for the target distribution. As such, one should be cautious when using the ideas from this paper in settings where humans are impacted and our assumptions cannot be verified. Indeed, biases present in training data could lead to unwanted biases in the resulting model, and our results do not have known provable guarantees when training complex neural networks as is often done in practice. However, we hope the ideas presented in this paper, and in particular, the ideas of model reuse for efficient mixture search, serve as useful starting points for deploying such models in real-world settings.

## Acknowledgments and Disclosure of Funding

This work was partially supported by NSF Grants 1704778, 1646522, and 1609279, ARO Grant W911NF-17-1-0359, a research gift from Qualcomm, the NXP Fellowship and the WNCG Industrial Affiliates Program.

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
