[Supplementary Material]

# A A More Detailed Discussion on Prior Work

Transfer learning has assumed an increasingly important role, especially in settings where we are either computationally limited, or data-limited, and yet we have the opportunity to leverage significant computational and data resources yet on domains that differ slightly from the target domain [30, 29, 10]. This has become an important paradigm in neural networks and other areas [39, 28, 5, 22].

An important related problem is that of covariate shift [36, 40, 13]. The problem here is that the target distribution may be different from the training distribution. A common technique for addressing this problem is by reweighting the samples in the training set, so that the distribution better matches that of the training set. There have been a number of techniques for doing this. An important recent thread has attempted to do this by using unlabelled data [20, 13]. Other approaches have considered a related problem of solving a weighted log-likelihood maximization [36], or by some form of importance sampling [37, 38] or bias correction [40]. In [25], the authors study a related problem of learning from different datasets, but provide mini-max bounds in terms of an agnostically chosen test distribution.

Our work is related to, but differs from all the above. As we explain in Section 3, we share the goal of transfer learning: we have access to enough data for training, but from a family of distributions that are different than the validation distribution (from which we have only enough data to validate). Under a model of covariate shift due to unobserved variables, we show that a target goal is finding an optimal reweighting of populations rather than data points. We use optimistic tree search to address precisely this problem – something that, as far as we know, has not been undertaken.

A key part of our work is working under a computational budget, and then designing an optimistic tree-search algorithm under uncertainty. We use a single SGD iteration as the currency denomination of our budget – i.e., our computational budget requires us to minimize the number of SGD steps in total that our algorithm computes. Enabling MCTS requires a careful understanding of SGD dynamics, and the error bounds on early stopping. There have been important SGD results studying early stopping, e.g., [17, 7] and generally results studying error rates for various versions of SGD and recentered SGD [27, 11, 32]. Our work requires a new high probability bound, which we obtain in the Supplemental material, Section D. In [27], the authors have argued that a uniform norm bound on the stochastic gradients is not the best assumption, however the results in that paper are in expectation. In this paper, we derive our SGD high-probability bounds under the mild assumption that the SGD gradient norms are bounded only at the optimal weight $w^*$.

There are several papers [18, 31] which derive high probability bounds on the suffix averaged and final iterates returned by SGD for non-smooth strongly convex functions. However, both papers operate under the assumption of uniform bounds on the stochastic gradient. Although these papers do not directly report a dependence on the diameter of the space, since they both consider projected gradient descent, one could easily translate their constant dependence to a sum of a diameter dependent term and a stochastic noise term (by using the bounded gradient assumption from [27], for example). However, as the set into which the algorithm would project is unknown to our algorithm (i.e., it would require knowing $w^*$), we cannot use projected gradient descent in our analysis. As we see in later sections, we need a high-probability SGD guarantee which characterizes the dependence on diameter of the space and noise of the stochastic gradient. It is not immediately clear how the analysis in [18, 31] could be extended in this setting under the gradient bounded assumption in [27]. In Appendix D, we instead develop the high probability bounds that are needed in our setting.

Optimistic tree search makes up the final important ingredient in our algorithm. These ideas have been used in a number of settings [9, 14]. Most relevant to us is a recent extension of these ideas to a setting with biased search [34, 35].

# B Standard Definitions from Convex Optimization

Recall that we assume throughout the paper that our loss functions satisfy the following assumptions similar to [27]:

**Assumption 1** (Restated from main text). *For each loss function $f(\cdot; z)$ corresponding to a **sample** $z \in \mathcal{Z}$, we assume that $f(\cdot; z)$ is:* (i) *$\beta$-**smooth** (Definition Definition 2) and* (ii) ***convex** (Definition 3).*

*Additionally, we assume that, for each $\boldsymbol{\alpha} \in \triangle$, the **averaged** loss function $F^{(\boldsymbol{\alpha})}(\cdot)$ is:* (i) $\mu$-***strongly convex** (Definition 4).*

We now state the definitions of these notions, which are standard in the optimization literature (see, for example, [8]).

**Definition 2** ($\beta$-smooth)**.** *We call a function $g(\cdot)$ $\beta$-smooth when, for all $\boldsymbol{w}, \boldsymbol{w}' \in \mathcal{W}$ when the gradient of $f$ is $\beta$-Lipschitz, i.e.,*

$$\|\nabla g(\boldsymbol{w}) - \nabla g(\boldsymbol{w}')\|_2 \leq \beta \|\boldsymbol{w} - \boldsymbol{w}'\|_2.$$

**Definition 3** (Convex)**.** *We call a function $g(\cdot)$ convex when, for all $\boldsymbol{w}, \boldsymbol{w}' \in \mathcal{W}$,*

$$g(\boldsymbol{w}) \geq g(\boldsymbol{w}') + \langle \nabla g(\boldsymbol{w}'), \boldsymbol{w} - \boldsymbol{w}' \rangle.$$

**Definition 4** ($\mu$-strongly convex)**.** *We call a function $g(\cdot)$ $\mu$-strongly convex if, for all $\boldsymbol{w}, \boldsymbol{w}' \in \mathcal{W}$,*

$$g(\boldsymbol{w}) \geq g(\boldsymbol{w}') + \langle \nabla g(\boldsymbol{w}'), \boldsymbol{w} - \boldsymbol{w}' \rangle + \frac{\mu}{2} \|\boldsymbol{w} - \boldsymbol{w}'\|_2^2.$$

## C  Smoothness with Respect to $\boldsymbol{\alpha}$

### C.1  Statement of main results

Our first result shows that the optimal weights with respect to the two distributions $p^{(\boldsymbol{\alpha}_1)}$ and $p^{(\boldsymbol{\alpha}_2)}$ are close, if the mixture weights $\boldsymbol{\alpha}_1$ and $\boldsymbol{\alpha}_2$ are close. This is the crucial observation upon which Corollary 2 relies.

**Theorem 4** (Restatement of Theorem 1)**.** *Consider a loss function $f(\boldsymbol{w}; z)$ which satisfies Assumption 1 and Assumption 2, and a convex body $\mathcal{X} = \mathrm{Conv}\{\boldsymbol{w}^*(\boldsymbol{\alpha}) \in \mathcal{W} \mid \boldsymbol{\alpha} \in \mathcal{A}\}$. Then for any $\boldsymbol{\alpha}_1, \boldsymbol{\alpha}_2 \in \triangle$, $\|\boldsymbol{w}^*(\boldsymbol{\alpha}_1) - \boldsymbol{w}^*(\boldsymbol{\alpha}_2)\|_2 \leq \frac{2\sigma \|\boldsymbol{\alpha}_1 - \boldsymbol{\alpha}_2\|_1}{\mu}$. where $\sigma^2 = \sup_{\boldsymbol{w}, \boldsymbol{w}' \in \mathcal{X}} \sup_{\boldsymbol{\alpha} \in \mathcal{A}} \beta^2 \|\boldsymbol{w} - \boldsymbol{w}'\|^2 + \mathcal{G}_*(\boldsymbol{\alpha})$.*

The above theorem is essentially a generalization of Theorem 3.9 in [17] to the case when only $\mathbb{E}[f]$, not $f$, is strongly convex. Theorem 4 implies that, if the partitions are such that for any cell $(h, i)$ at height $h$, $\|\boldsymbol{\alpha}_1 - \boldsymbol{\alpha}_2\|_1 \leq \nu' \rho^h$ for all $\boldsymbol{\alpha}_1, \boldsymbol{\alpha}_2 \in (h, i)$, where $\rho \in (0, 1)$, then we have that $\|\boldsymbol{w}^*(\boldsymbol{\alpha}_1) - \boldsymbol{w}^*(\boldsymbol{\alpha}_2)\|_2 \leq \nu_1 \rho^h$, for some $\nu_1 \geq 0$. We note that such a partitioning does indeed exist:

**Corollary 2** (of Theorem 4, restatement of Corollary 1)**.** *There exists a hierarchical partitioning $\mathcal{P}$ of the simplex of mixture weights $\mathcal{A}$ (namely, the simplex bisection strategy described in [21]) such that, for any cell $(h, i) \in \mathcal{P}$, and any $\boldsymbol{\alpha}_1, \boldsymbol{\alpha}_2 \in (h, i)$,*

$$\|\boldsymbol{\alpha}_1 - \boldsymbol{\alpha}_2\|_1 \leq \sqrt{2K} \left(\frac{\sqrt{3}}{2}\right)^{\frac{h}{K-1}-1}, \tag{6}$$

*where $K - 1 = \dim(\triangle)$. Combined with Theorem 4, this implies*

$$\|\boldsymbol{w}^*(\boldsymbol{\alpha}_1) - \boldsymbol{w}^*(\boldsymbol{\alpha}_2)\|_2^2 \leq \nu_1 \rho^h \tag{7}$$

*and*

$$|G(\boldsymbol{\alpha}_1) - G(\boldsymbol{\alpha}_2)| \leq \nu_2 \rho_2^h, \tag{8}$$

*where $\nu_1 = \left(\frac{4\sigma\sqrt{2K}}{\sqrt{3}\mu}\right)^2$, $\rho = \left(\frac{\sqrt{3}}{2}\right)^{\frac{2}{K-1}}$, $\nu_2 = L\sqrt{\nu_1}$, $L = \beta \sup_{\boldsymbol{\alpha} \in \triangle} \|\boldsymbol{w}^*(\boldsymbol{\alpha}) - \boldsymbol{w}^*(\boldsymbol{\alpha}^*)\|$, and $\rho_2 = \sqrt{\rho}$.*

Refer to Appendix C for the proofs of these claims.

### C.2  Proving the main results

In this section we prove Theorem 4. The analysis is an interesting generalization of Theorem 3.9 in [17]. The key technique is to create a total variational coupling between $\boldsymbol{\alpha}_1$ and $\boldsymbol{\alpha}_2$. Then using this coupling we prove that SGD iterates from the two distributions cannot be too far apart in expectation. Therefore, because the two sets of iterates converge to their respective optimal solutions, we can conclude that the optimal weights $\boldsymbol{w}^*(\boldsymbol{\alpha}_1)$ and $\boldsymbol{w}^*(\boldsymbol{\alpha}_2)$ are close.

**Lemma 1.** *Under conditions of Theorem 4, let $\boldsymbol{w}_n(\boldsymbol{\alpha}_1)$ and $\boldsymbol{w}_n(\boldsymbol{\alpha}_2)$ be the random variables representing the weights after performing $n$ steps of online projected SGD onto a convex body $\mathcal{X} = \mathrm{Conv}\{\boldsymbol{w}^*(\boldsymbol{\alpha}) \mid \boldsymbol{\alpha} \in \mathcal{A}\}$ using the data distributions represented by the mixtures $\boldsymbol{\alpha}_1$ and $\boldsymbol{\alpha}_2$ respectively, starting from the same initial weight $\boldsymbol{w}_0$, and using the step size sequence*

$$\eta_t = \frac{2}{\mu\left(t + \frac{1}{\mu}\max\{\kappa^2(\mu+\beta), 4\beta\}\right)}$$

*Then we have the following bound,*

$$\mathbb{E}\left[\|\boldsymbol{w}_n(\boldsymbol{\alpha}_1) - \boldsymbol{w}_n(\boldsymbol{\alpha}_2)\|\right] \leq \frac{2\sigma\|\boldsymbol{\alpha}_1 - \boldsymbol{\alpha}_2\|_1}{\mu}.$$

*where $\sigma^2 = \sup_{\boldsymbol{w},\boldsymbol{w}'\in\mathcal{X}} \sup_{\boldsymbol{\alpha}\in\mathcal{A}} \beta^2\|\boldsymbol{w}-\boldsymbol{w}'\|^2 + \mathcal{G}_*(\boldsymbol{\alpha})$.*

*Proof.* We closely follow the proof of Theorem 3.9 in [17]. Let $\boldsymbol{w}_{t+1}(\boldsymbol{\alpha}_i) = \Pi_{\mathcal{X}}(\boldsymbol{w}_t - \eta_t\nabla f(\boldsymbol{w}_t; Z_t^{(i)}))$ denote the SGD update while processing the $t$-th example from $\boldsymbol{\alpha}_i$ for $i \in \{1,2\}$. Let $I, J$ be two random variables whose joint distribution follows the variational coupling between $\boldsymbol{\alpha}_1$ and $\boldsymbol{\alpha}_2$. Thus the marginals of $I$ and $J$ are $\boldsymbol{\alpha}_1$ and $\boldsymbol{\alpha}_2$ respectively, while $\mathbb{P}(I \neq J) = d_{TV}(\boldsymbol{\alpha}_1, \boldsymbol{\alpha}_2)$. At each time $I_t \sim I$ and $J_t \sim J$ are drawn. If $I_t = J_t$, then we draw a data sample $Z_t$ from $D_{I_t}$ and set $Z_t^{(1)} = Z_t^{(2)} = Z_t$. Otherwise, we draw $Z_t^{(1)}$ from $D_{I_t}$ and $Z_t^{(2)}$ from $D_{J_t}$ independently.

Therefore, following the analysis in [17], if $I_t = J_t$, then, by Lemma 3.7.3 in [17], by our choice of step size, and since Euclidean projection does not increase the distance between projected points (see for example Lemma 3.1 in [8]),

$$\begin{aligned}
\delta_{t+1}^2 &= \|\boldsymbol{w}_{t+1}(\boldsymbol{\alpha}_1) - \boldsymbol{w}_{t+1}(\boldsymbol{\alpha}_2)\|^2 \\
&= \|\Pi_{\mathcal{X}}(\boldsymbol{w}_t(\boldsymbol{\alpha}_1) - \eta_t\nabla f(\boldsymbol{w}_t(\boldsymbol{\alpha}_1); Z_t)) - \Pi_{\mathcal{X}}(\boldsymbol{w}_t(\boldsymbol{\alpha}_2) - \eta_t\nabla f(\boldsymbol{w}_t(\boldsymbol{\alpha}_2); Z_t))\|^2 \\
&\leq \|\boldsymbol{w}_t(\boldsymbol{\alpha}_1) - \eta_t\nabla f(\boldsymbol{w}_t(\boldsymbol{\alpha}_1); Z_t) - \boldsymbol{w}_t(\boldsymbol{\alpha}_2) + \eta_t\nabla f(\boldsymbol{w}_t(\boldsymbol{\alpha}_2; Z_t)\|^2 \\
&= \delta_t^2 + \eta_t^2\|\nabla f(\boldsymbol{w}_t(\boldsymbol{\alpha}_1); Z_t) - \nabla f(\boldsymbol{w}_t(\boldsymbol{\alpha}_2; Z_t)\|^2 \\
&\quad - 2\eta_t\langle\nabla f(\boldsymbol{w}_t(\boldsymbol{\alpha}_1); Z_t) - \nabla f(\boldsymbol{w}_t(\boldsymbol{\alpha}_2); Z_t), \boldsymbol{w}_t(\boldsymbol{\alpha}_1) - \boldsymbol{w}_t(\boldsymbol{\alpha}_2)\rangle
\end{aligned}$$

Now, taking expectations with respect to $Z_t$, we get the following:

$$\mathbb{E}_{Z_t}[\mathbb{1}\{I_t = J_t\}\delta_{t+1}^2]$$

$$\leq \mathbb{P}(I_t = J_t)\delta_t^2 + \eta_t^2 \mathbb{E}_{Z_t}\mathbb{1}\{I_t = J_t\}\underbrace{\|\nabla f(\boldsymbol{w}_t(\boldsymbol{\alpha}_1); Z_t) - \nabla f(\boldsymbol{w}_t(\boldsymbol{\alpha}_2); Z_t)\|^2}_{\leq \beta^2 \delta_t^2 \text{ by smoothness of } f}$$

$$- 2\eta_t \mathbb{E}_{Z_t}\mathbb{1}\{I_t = J_t\}\langle \nabla f(\boldsymbol{w}_t(\boldsymbol{\alpha}_1); Z_t) - \nabla f(\boldsymbol{w}_t(\boldsymbol{\alpha}_2); Z_t), \boldsymbol{w}_t(\boldsymbol{\alpha}_1) - \boldsymbol{w}_t(\boldsymbol{\alpha}_2)\rangle$$

$$\leq \mathbb{P}(I_t = J_t)\delta_t^2 + \mathbb{P}(I_t = J_t)\eta_t^2 \beta^2 \delta_t^2$$

$$- 2\eta_t \sum_{i=1}^{K} \mathbb{E}_{Z_t}\left[\left\langle \nabla f(\boldsymbol{w}_t(\boldsymbol{\alpha}_1); Z_t) - \nabla f(\boldsymbol{w}_t(\boldsymbol{\alpha}_2); Z_t),\right.\right.$$

$$\left.\left. \boldsymbol{w}_t(\boldsymbol{\alpha}_1) - \boldsymbol{w}_t(\boldsymbol{\alpha}_2)\right\rangle \mid I_t = i = J_t\right]\mathbb{P}(I_t = i = J_t)$$

$$\overset{(a)}{=} \mathbb{P}(I_t = J_t)\delta_t^2 + \mathbb{P}(I_t = J_t)\eta_t^2 \beta^2 \delta_t^2$$

$$- 2\eta_t \sum_{i=1}^{K} \langle \nabla F^{(e_i)}(\boldsymbol{w}_t(\boldsymbol{\alpha}_1)) - \nabla F^{(e_i)}(\boldsymbol{w}_t(\boldsymbol{\alpha}_2)), \boldsymbol{w}_t(\boldsymbol{\alpha}_1) - \boldsymbol{w}_t(\boldsymbol{\alpha}_2)\rangle \mathbb{P}(I_t = i = J_t)$$

$$\leq \mathbb{P}(I_t = J_t)\delta_t^2 + \mathbb{P}(I_t = J_t)\eta_t^2 \beta^2 \delta_t^2$$

$$- 2\eta_t \mathbb{P}(I_t = J_t)$$

$$\times \underbrace{\left(\frac{\mu\beta}{\mu+\beta}\|\boldsymbol{w}_t(\boldsymbol{\alpha}_1) - \boldsymbol{w}_t(\boldsymbol{\alpha}_2)\|^2 + \frac{1}{\mu+\beta}\underbrace{\|\nabla F^{(e_1)}(\boldsymbol{w}_t(\boldsymbol{\alpha}_1)) - \nabla F^{(e_1)}(\boldsymbol{w}_t(\boldsymbol{\alpha}_2))\|^2}_{\leq \mu^2 \delta_t^2 \text{ by strong convexity of } F}\right)}_{\text{bound holds by Lemma 3.11 in [8]}}$$

$$= \mathbb{P}(I_t = J_t)\left(\left(1 - 2\eta_t\frac{\mu\beta}{\mu+\beta}\right)\delta_t^2 - \eta_t\underbrace{\left(\frac{2\mu^2}{\mu+\beta} - \beta^2\eta_t\right)}_{\geq 0 \text{ by choice of } \eta_t}\delta_t^2\right)$$

$$\overset{(b)}{\leq} \mathbb{P}(I_t = J_t)(1 - \mu\eta_t)\delta_t^2$$

where $(a)$ follows from linearity of expectation, and noting that, conditioned on the event $I_t = i$, $Z_t \sim \mathcal{D}_i$. The last inequality $(b)$ follows since $\frac{2\beta}{\mu+\beta} \geq 1$ (since $\beta \geq \mu$), and since $\eta_t = \frac{2}{\mu\left(t + \frac{1}{\mu}\max\{\kappa^2(\mu+\beta), 4\beta\}\right)} \leq \frac{2}{\kappa^2(\mu+\beta)}$. Thus, when $I_t = J_t$, we have that

$$\mathbb{E}_{Z_t}[\mathbb{1}\{I_t = J_t\}\delta_{t+1}] \leq \sqrt{\mathbb{E}_{Z_t}[\mathbb{1}\{I_t = J_t\}\delta_{t+1}^2]} \qquad \text{Using concavity of } \sqrt{\cdot} \text{ and Jensen's}$$

$$\leq \sqrt{\mathbb{P}(I_t = J_t)(1 - \mu\eta_t)\delta_t^2} \qquad \text{using our bound above}$$

$$\leq \mathbb{P}(I_t = J_t)(1 - \mu\eta_t/2)\delta_t \qquad \text{since } \sqrt{1 - \mu\eta_t} \leq 1 - \frac{\mu\eta_t}{2}.$$

On the other hand, when $I_t \neq J_t$, we have that

$$\delta_{t+1} \leq \|\boldsymbol{w}_t(\boldsymbol{\alpha}_1) - \eta_t\nabla f(\boldsymbol{w}_t(\boldsymbol{\alpha}_1; Z_t^{(1)}) - \boldsymbol{w}_t(\boldsymbol{\alpha}_2) + \eta_t\nabla f(\boldsymbol{w}_t(\boldsymbol{\alpha}_2; Z_t^{(2)}))\|$$

$$\leq \|\boldsymbol{w}_t(\boldsymbol{\alpha}_1) - \eta_t\nabla f(\boldsymbol{w}_t(\boldsymbol{\alpha}_1; Z_t^{(1)}) - \boldsymbol{w}_t(\boldsymbol{\alpha}_2) + \eta_t\nabla f(\boldsymbol{w}_t(\boldsymbol{\alpha}_2; Z_t^{(1)}))\|$$

$$+ \eta_t(\|\nabla f(\boldsymbol{w}(\boldsymbol{\alpha}_2); Z^{(1)})\| + \|\nabla f(\boldsymbol{w}(\boldsymbol{\alpha}_2); Z^{(2)})\|)$$

$$\leq (1 - \mu\eta_t/2)\delta_t + \eta_t(\|\nabla f(\boldsymbol{w}(\boldsymbol{\alpha}_2); Z^{(1)})\| + \|\nabla f(\boldsymbol{w}(\boldsymbol{\alpha}_2); Z^{(2)})\|) \qquad \text{By above}$$

$$\leq (1 - \mu\eta_t/2)\delta_t + \eta_t\sqrt{2\beta^2\|\boldsymbol{w}_t(\boldsymbol{\alpha}_2) - \boldsymbol{w}^*(\boldsymbol{\alpha}_1)\|^2 + 2\mathcal{G}_*(\boldsymbol{\alpha}_1)}$$

$$+ \eta_t\sqrt{2\beta^2\|\boldsymbol{w}_t(\boldsymbol{\alpha}_2) - \boldsymbol{w}^*(\boldsymbol{\alpha}_2)\|^2 + 2\mathcal{G}_*(\boldsymbol{\alpha}_2))} \qquad \text{by Lemma 2}$$

$$\leq (1 - \mu\eta_t/2)\delta_t + 2\sigma\eta_t$$

where $\sigma^2 = \sup_{\boldsymbol{w}, \boldsymbol{w}' \in \mathcal{X}} \sup_{\boldsymbol{\alpha} \in \mathcal{A}} \beta^2 \|\boldsymbol{w} - \boldsymbol{w}'\|^2 + \mathcal{G}_*(\boldsymbol{\alpha})$.

Thus, by combining both of these results, we obtain:

$$\mathbb{E}[\delta_{t+1}] \leq (1 - \mu\eta_t/2)\mathbb{E}[\delta_t] + 2\sigma\eta_t \mathbb{P}\{I_t \neq J_t\}$$
$$= (1 - \mu\eta_t/2)\mathbb{E}[\delta_t] + \sigma\eta_t \|\boldsymbol{\alpha}_1 - \boldsymbol{\alpha}_2\|_1.$$

Since by construction, $\delta_0 = 0$, we get the following result from the recursion,

$$\mathbb{E}[\delta_n] \leq \sum_{t=t_0}^{n} \left\{ \prod_{s=t+1}^{n} \left( 1 - \frac{1}{s+E} \right) \right\} \frac{2\sigma}{\mu(t+E)} \|\boldsymbol{\alpha}_1 - \boldsymbol{\alpha}_2\|_1$$
$$= \sum_{t=t_0}^{n} \frac{t+E}{n+E} \frac{2\sigma}{\mu(t+E)} \|\boldsymbol{\alpha}_1 - \boldsymbol{\alpha}_2\|_1$$
$$\leq \frac{n-t_0+1}{n+E} \frac{2\sigma}{\mu} \|\boldsymbol{\alpha}_1 - \boldsymbol{\alpha}_2\|_1$$
$$\leq \frac{2\sigma}{\mu} \|\boldsymbol{\alpha}_1 - \boldsymbol{\alpha}_2\|_1.$$

$\square$

*Proof of Theorem 4.* First, note that by definition $\boldsymbol{w}^*(\boldsymbol{\alpha})$ is not a random variable i.e it is the optimal weight with respect to the distribution corresponding to $\boldsymbol{\alpha}$. On the other hand, $\boldsymbol{w}_n(\cdot)$ is a random variable, where the randomness is coming from the randomness in SGD sampling. By the triangle inequality, we have the following:

$$\|\boldsymbol{w}^*(\boldsymbol{\alpha}_1) - \boldsymbol{w}^*(\boldsymbol{\alpha}_2)\| \leq \|\boldsymbol{w}^*(\boldsymbol{\alpha}_1) - \boldsymbol{w}_n(\boldsymbol{\alpha}_1)\| + \|\boldsymbol{w}_n(\boldsymbol{\alpha}_1) - \boldsymbol{w}_n(\boldsymbol{\alpha}_2)\| +$$
$$\|\boldsymbol{w}^*(\boldsymbol{\alpha}_2) - \boldsymbol{w}_n(\boldsymbol{\alpha}_2)\|$$
$$\implies \|\boldsymbol{w}^*(\boldsymbol{\alpha}_1) - \boldsymbol{w}^*(\boldsymbol{\alpha}_2)\| = \mathbb{E}[\|\boldsymbol{w}^*(\boldsymbol{\alpha}_1) - \boldsymbol{w}^*(\boldsymbol{\alpha}_2)\|]$$
$$\leq \mathbb{E}[\|\boldsymbol{w}^*(\boldsymbol{\alpha}_1) - \boldsymbol{w}_n(\boldsymbol{\alpha}_1)\|] + \mathbb{E}[\|\boldsymbol{w}_n(\boldsymbol{\alpha}_1) - \boldsymbol{w}_n(\boldsymbol{\alpha}_2)\|]$$
$$+ \mathbb{E}[\|\boldsymbol{w}^*(\boldsymbol{\alpha}_2) - \boldsymbol{w}_n(\boldsymbol{\alpha}_2)\|]. \tag{9}$$

The expectation in the middle of the r.h.s. is bounded as in Lemma 1. The other two terms, on the other hand, can be made *arbitrarily* small by choosing $n$ sufficiently large. Indeed, using the step size schedule as in Lemma 1, we can use Theorem 2 in [27][1] [2] and Jensen's inequality to bound the other two terms on the r.h.s. as

$$\mathbb{E}[\|\boldsymbol{w}_n(\boldsymbol{\alpha}_1) - \boldsymbol{w}^*(\boldsymbol{\alpha}_1)\|_2] \leq \sqrt{\mathbb{E}[\|\boldsymbol{w}_n(\boldsymbol{\alpha}_1) - \boldsymbol{w}^*(\boldsymbol{\alpha}_1)\|_2^2]} \qquad \text{by concavity of } \sqrt{\cdot}.$$
$$\leq \sqrt{\frac{32\mathcal{G}_*}{\mu^2(n-T+E)}}$$

where we take $\mathcal{G}_* = \max\{\mathcal{G}_*(\boldsymbol{\alpha}_1), \mathcal{G}_*(\boldsymbol{\alpha}_2)\}$, $E = \frac{1}{\mu}\max\{\kappa^2(\mu + \beta), 4\beta\}$, $T = 4\kappa \max\left\{ \frac{\beta\mu}{\mathcal{G}_*} \|\boldsymbol{w}_0(\boldsymbol{\alpha}_1) - \boldsymbol{w}^*(\boldsymbol{\alpha}_1)\|^2, 1 \right\} - 4\kappa$, and as long as $n \geq T$. Now, noting that the inequality Equation 9 holds for all $n$, we have the bound claimed in Theorem 4. $\square$

*Proof of Corollary 2.* This proof is a straightforward consequence of Theorem 3.1 in [21] and Theorem 4. In particular, Theorem 3.1 in [21] tells us that under the method of bisection of the simplex which they describe,

$$\|\boldsymbol{\alpha}_1 - \boldsymbol{\alpha}_2\|_2 \leq \left( \frac{\sqrt{3}}{2} \right)^{\lfloor \frac{h}{K-1} \rfloor} \text{diam}(\triangle),$$

where $\mathrm{diam}(\triangle) = \sup\{\|\boldsymbol{\alpha} - \boldsymbol{\alpha}'\|_2 \mid \boldsymbol{\alpha}, \boldsymbol{\alpha}' \in \triangle\}$, and $K - 1 = \dim(\triangle)$. As noted in Remark 2.5 in [21], $\mathrm{diam}(\triangle) = \sqrt{2}$ since $\triangle$ is the unit simplex. Thus, by the Cauchy-Schwartz inequality, and since $\left\lfloor \frac{h}{K-1} \right\rfloor > \frac{h}{K-1} - 1$, we have the following:

$$\|\boldsymbol{\alpha}_1 - \boldsymbol{\alpha}_2\|_1 \leq \sqrt{K}\|\boldsymbol{\alpha}_1 - \boldsymbol{\alpha}_2\|_2$$
$$\leq \sqrt{2K}\left(\frac{\sqrt{3}}{2}\right)^{\lfloor \frac{h}{K-1} \rfloor}$$
$$\leq \sqrt{2K}\left(\frac{\sqrt{3}}{2}\right)^{\frac{h}{K-1} - 1}.$$

Now, from Assumption 1 and Cauchy-Schwarz, we have that

$$F^{(\boldsymbol{\alpha}^*)}(\boldsymbol{w}^*(\boldsymbol{\alpha}_1)) - F^{(\boldsymbol{\alpha}^*)}(\boldsymbol{w}^*(\boldsymbol{\alpha}_2))$$
$$\geq \langle \nabla F^{(\boldsymbol{\alpha}^*)}(\boldsymbol{w}^*(\boldsymbol{\alpha}_2)), \boldsymbol{w}^*(\boldsymbol{\alpha}_1) - \boldsymbol{w}^*(\boldsymbol{\alpha}_2)\rangle + \frac{\mu}{2}\|\boldsymbol{w}^*(\boldsymbol{\alpha}_1) - \boldsymbol{w}^*(\boldsymbol{\alpha}_2)\|^2$$
$$\geq \left(-\|\nabla F^{(\boldsymbol{\alpha}^*)}(\boldsymbol{w}^*(\boldsymbol{\alpha}_2))\| + \frac{\mu}{2}\|\boldsymbol{w}^*(\boldsymbol{\alpha}_1) - \boldsymbol{w}^*(\boldsymbol{\alpha}_2)\|\right)\|\boldsymbol{w}^*(\boldsymbol{\alpha}_1) - \boldsymbol{w}^*(\boldsymbol{\alpha}_2)\|$$
$$= \left(-\|\nabla F^{(\boldsymbol{\alpha}^*)}(\boldsymbol{w}^*(\boldsymbol{\alpha}_2)) - \underbrace{\nabla F^{(\boldsymbol{\alpha}^*)}(\boldsymbol{w}^*(\boldsymbol{\alpha}^*))}_{=0}\| + \frac{\mu}{2}\|\boldsymbol{w}^*(\boldsymbol{\alpha}_1) - \boldsymbol{w}^*(\boldsymbol{\alpha}_2)\|\right)$$
$$\underbrace{\phantom{xxxxxxxxxxxxxxxxxxxxxxxxxxxxxxxxx}}_{\leq \beta\|\boldsymbol{w}^*(\boldsymbol{\alpha}_2) - \boldsymbol{w}^*(\boldsymbol{\alpha}^*)\|}$$
$$\times \|\boldsymbol{w}^*(\boldsymbol{\alpha}_1) - \boldsymbol{w}^*(\boldsymbol{\alpha}_2)\|$$
$$\geq \left(-\beta\|\boldsymbol{w}^*(\boldsymbol{\alpha}_2) - \boldsymbol{w}^*(\boldsymbol{\alpha}^*)\| + \frac{\mu}{2}\|\boldsymbol{w}^*(\boldsymbol{\alpha}_1) - \boldsymbol{w}^*(\boldsymbol{\alpha}_2)\|\right)\|\boldsymbol{w}^*(\boldsymbol{\alpha}_1) - \boldsymbol{w}^*(\boldsymbol{\alpha}_2)\|$$

and by a similar argument, exchanging the roles of $\boldsymbol{\alpha}_1$ and $\boldsymbol{\alpha}_2$, we have

$$F^{(\boldsymbol{\alpha}^*)}(\boldsymbol{w}^*(\boldsymbol{\alpha}_2)) - F^{(\boldsymbol{\alpha}^*)}(\boldsymbol{w}^*(\boldsymbol{\alpha}_1))$$
$$\geq \left(-\beta\|\boldsymbol{w}^*(\boldsymbol{\alpha}_1) - \boldsymbol{w}^*(\boldsymbol{\alpha}^*)\| + \frac{\mu}{2}\|\boldsymbol{w}^*(\boldsymbol{\alpha}_1) - \boldsymbol{w}^*(\boldsymbol{\alpha}_2)\|\right)\|\boldsymbol{w}^*(\boldsymbol{\alpha}_1) - \boldsymbol{w}^*(\boldsymbol{\alpha}_2)\|$$

Thus, we have that

$$|F^{(\boldsymbol{\alpha}^*)}(\boldsymbol{w}^*(\boldsymbol{\alpha}_1)) - F^{(\boldsymbol{\alpha}^*)}(\boldsymbol{w}^*(\boldsymbol{\alpha}_2))|$$
$$\leq \max_{\boldsymbol{\alpha} \in \{\boldsymbol{\alpha}_1, \boldsymbol{\alpha}_2\}} \left|\beta\|\boldsymbol{w}^*(\boldsymbol{\alpha}) - \boldsymbol{w}^*(\boldsymbol{\alpha}^*)\| - \frac{\mu}{2}\|\boldsymbol{w}^*(\boldsymbol{\alpha}_1) - \boldsymbol{w}^*(\boldsymbol{\alpha}_2)\|\right|\|\boldsymbol{w}^*(\boldsymbol{\alpha}_1) - \boldsymbol{w}^*(\boldsymbol{\alpha}_2)\|$$

Now, taking $L = \beta \sup_{\boldsymbol{\alpha} \in \triangle}\|\boldsymbol{w}^*(\boldsymbol{\alpha}) - \boldsymbol{w}^*(\boldsymbol{\alpha}^*)\|$, we may use this result along with Theorem 4 to obtain:

$$|G(\boldsymbol{\alpha}_1) - G(\boldsymbol{\alpha}_2)| = |F^{(te)}(\boldsymbol{w}^*(\boldsymbol{\alpha}_1)) - F^{(te)}(\boldsymbol{w}^*(\boldsymbol{\alpha}_2))|$$
$$= |F^{(\boldsymbol{\alpha}^*)}(\boldsymbol{w}^*(\boldsymbol{\alpha}_1)) - F^{(\boldsymbol{\alpha}^*)}(\boldsymbol{w}^*(\boldsymbol{\alpha}_2))|$$
$$\leq L\|\boldsymbol{w}^*(\boldsymbol{\alpha}_1) - \boldsymbol{w}^*(\boldsymbol{\alpha}_2)\|_2$$
$$\leq \frac{2L\sigma\|\boldsymbol{\alpha}_1 - \boldsymbol{\alpha}_2\|_1}{\mu}$$
$$\leq \frac{4L\sigma\sqrt{2K}}{\sqrt{3}\mu}\left(\frac{\sqrt{3}}{2}\right)^{\frac{h}{K-1}},$$

which is the desired result. $\qquad\square$

## D  New High-Probability Bounds on SGD without a Constant Gradient Bound

In this section, we will prove a high-probability bound on any iterate of SGD evolving over the time interval $t = 1, 2, \ldots, T$, without assuming a uniform bound on the stochastic gradient over the

domain. Instead, this bound introduces a tunable parameter $\Lambda > (T+1)$ that controls the trade-off between a bound on the SGD iterate $d_t^2$, and the probability with which the bound holds. As we discuss in Remark 3, this parameter can be set to provide tighter high-probability guarantees on the SGD iterates in settings where the diameter of the domain is large and/or cannot be controlled.

**Theorem 5** (Formal statement of Theorem 2). *Consider a sequence of random samples $z_0, z_1, \ldots, z_T$ drawn from a distribution $p(z)$. Define the filtration $\mathcal{F}_t$ generated by $\sigma\{z_0, z_1, \ldots, z_t\}$. Let us define a sequence of random variables by the gradient descent update: $\boldsymbol{w}_{t+1} = \boldsymbol{w}_t - \eta_t \nabla f(\boldsymbol{w}_t; z_t), t = 1, \ldots, T$, and $\boldsymbol{w}_0$ is a fixed vector in $\mathcal{W}$. Take $d_t^2 = \|\boldsymbol{w}_t - \boldsymbol{w}^*\|_2^2$.*

*If we use the step size schedule $\eta_t = \frac{2}{\mu(t+E)}$, where $E = 4096\kappa^2 \log \Lambda^8$, then, under Assumption 1 and Assumption 2, and taking $\Lambda \geq t+1$, we have the following high probability bound on the final iterate of the SGD procedure after $t$ time steps for **any** $k \geq 0$:*

$$\Pr\left(d_{t+1}^2 > \frac{G(d_0^2, \mathcal{G}_*)}{t+E+1} + \frac{8(t+1)\tilde{C}(D^2, D\sqrt{\mathcal{G}_*})}{\mu(t+1+E)\Lambda^7} + \frac{4\sqrt{2\log(\Lambda^8)}\sqrt{\hat{C}(k; d_0, \mathcal{G}_*)}}{\mu(t+E+1)^{\alpha_{k+1}}}\right) \leq \frac{t+1}{\Lambda^8}$$

(10)

*where*

$$G(d_0^2, \mathcal{G}_*) = \max\left\{Ed_0^2, \frac{8\mathcal{G}_*}{\mu^2}\right\}$$

$$\tilde{C}(D^2, D\sqrt{\mathcal{G}_*}) = D\sqrt{8\beta^2 D^2 + 2\mathcal{G}_*}$$

$$\hat{C}(k; d_0, \mathcal{G}_*) = \hat{C}(k) = O(\log \Lambda^8) \qquad \text{Defintion in Corollary 4, discussed in Remark 4.}$$

$$\alpha_{k+1} = \sum_{i=1}^{k+1} \frac{1}{2^i}.$$

*In particular, when we choose $k = 0$, the above expression becomes*

$$\Pr\left(d_{t+1}^2 > \frac{G(d_0^2, \mathcal{G}_*)}{t+E+1} + \frac{8(t+1)\tilde{C}(D^2, D\sqrt{\mathcal{G}_*})}{\mu(t+1+E)\Lambda^7} + \frac{4\sqrt{2\check{C}}\log(\Lambda^8)}{\mu\sqrt{t+E+1}}\right) \leq \frac{t+1}{\Lambda^8} \qquad (11)$$

*where*

$$\check{C} = \max\left\{\frac{8d_0^2(4\beta^2 d_0^2 + \mathcal{G}_*)}{(1+E)\log \Lambda^8}, \left(\frac{32\sqrt{2}\mathcal{G}_*}{\mu} + \frac{2}{E}\right)^2, \left(\frac{64\beta^2 c_1^2}{(1+E)\log \Lambda^8} + \frac{8\mathcal{G}_* c_1}{\log \Lambda^8}\right)^2\right\}$$

$$c_1 = G(d_0^2, \mathcal{G}_*) + \frac{8\tilde{C}(D^2, D\sqrt{\mathcal{G}_*})}{\mu\Lambda^6}$$

**Remark 2.** *This result essentially states that the distance of $\boldsymbol{w}_t$ to $\boldsymbol{w}^*$ is at most the sum of three terms with high probability. Recall from the first step of the proof of Theorem 2 in [27] that $\mathbb{E}[d_t^2] \leq (1 - \mu\eta_t)\mathbb{E}[d_t^2] + 2\eta_t^2 \mathcal{G}_* + \mathbb{E}[M_t]$, where $M_t = \langle \nabla F(\boldsymbol{w}_t) - \nabla f(\boldsymbol{w}_t; z_t), \boldsymbol{w}_t - \boldsymbol{w}^*\rangle$ is a martingale difference sequence with respect to the filtration generated by samples $\boldsymbol{w}_0, \ldots, \boldsymbol{w}_t$ (in particular, note that $\mathbb{E}[M_t] = 0$). We obtain a similar inequality in the high probability analysis without the expectations, so bounding the $M_t$ term is the main difficulty in proving the high probability convergence guarantee. Indeed, the first term in our high-probability guarantee corresponds to a bound on the $(1 - \mu\eta_t)d_t + 2\eta_t \mathcal{G}_*$ term. Thus, as in the expected value analysis from [27], this term decreases linearly in the number of steps $t$, with the scaling constant depending only on the initial distance $d_0$ and a uniform bound on the stochastic gradient at the optimum model parameter ($\boldsymbol{w}^*$).*

*The latter two terms correspond to a bound on a normalized version of the martingale $\sum_i M_i$, which appears after unrolling the aforementioned recursion. Due to our more relaxed assumption on the bound on the norm of the stochastic gradient, we employ different techniques in bounding this term than were used in [18]. The second term is a bias term that depends on the worst case diameter bound $D$ (or if no diameter bound exists, then $D$ represents the worst case distance between $\boldsymbol{w}_t$ and $\boldsymbol{w}^*$, see Remark 3), and appears as a result of applying Azuma-Hoeffding with conditioning. Our bound exhibits a trade-off between the bias term which is $O(D^2/\text{poly}(\Lambda))$, and the probability of the*

*bad event which is $\frac{t+1}{\Lambda^8}$. This trade-off can be achieved by tuning the parameter $\Lambda$. Notice that while the probability of the bad event decays polynomially in $\Lambda$, the bias only increases as $\mathrm{poly}(\log \Lambda)$.*

*The third term represents the deviation of the martingale, which decreases nearly linearly in $t$ (i.e. $t^\gamma$ for any $\gamma$ close to 1). The scaling constant, however, depends on $\gamma$. By choosing $\Lambda$ appropriately (in the second term), this third term decays the slowest of the three, for large values of $t$, and is thus the most important one from a scaling-in-time perspective.*

**Remark 3.** *In typical SDG analysis (e.g. [12, 18]), a uniform bound on the stochastic gradient is assumed. Note that if we assume a uniform bound on $d_t$, i.e. $d_t \leq D \ \forall\ t \in [1, T]$, then under Assumption 1, we immediately obtain a uniform bound on the stochastic gradient, since:*

$$\|\nabla f(\boldsymbol{w}_t; z)\| \leq \|\nabla f(\boldsymbol{w}_t; z) - \nabla f(\boldsymbol{w}^*; z)\| + \|\nabla f(\boldsymbol{w}^*; z)\|$$
$$\leq \beta d_t + \sqrt{\mathcal{G}_*}$$
$$\leq \beta D + \sqrt{\mathcal{G}_*} := \sqrt{\bar{\mathcal{G}}} \tag{12}$$

*If we do not have access to a projection operator on our feasible set of $\boldsymbol{w}$, or otherwise choose not to run projected gradient descent, then we obtain a worst-case upper bound of $D = O(t^u)$ where $u = 2\sqrt{2}\kappa^{3/2}$, since:*

$$d_{t+1} \leq d_t + \eta_t \|\nabla f(\boldsymbol{w}_t; z_t)\| \qquad \text{by triangle inequality and definition of the SGD step}$$

$$\leq d_t + \eta_t \sqrt{2\beta^2 \kappa d_t^2 + 2\mathcal{G}_*} \qquad \text{by Lemma 2}$$

$$\leq \left(1 + \frac{\alpha\sqrt{2\kappa}\beta}{\mu(t+E)}\right) d_t + \frac{\alpha\sqrt{2\mathcal{G}_*}}{\mu(t+E)} \qquad \text{by choice of } \eta_t = \frac{\alpha}{\mu(t+E)}, \text{ where } \alpha > 1 \text{ must hold}$$

$$= O\left(t^{\alpha\sqrt{2}\kappa^{3/2}}\right) \qquad \text{we take } \alpha = 2 \text{ throughout this paper}$$

*Thus, when we do not assume access to the feasible set of $\boldsymbol{w}$ and do not run projected gradient descent, a convergence guarantee of the form $\tilde{O}\left(\frac{\bar{\mathcal{G}}}{t}\right)$ that follows from a uniform bound on the stochastic gradient does not suffice in our setting because $\bar{\mathcal{G}}$ scales polynomially in t. We further note that even if we do have access to a projection operator, $\bar{\mathcal{G}}$ scales quadratically in the radius of the projection set, and thus can be very large.*

*Instead, we wish to construct a high probability guarantee on the final SGD iterate in a fashion similar to the expected value guarantee given in [27]. Now under our construction, we have an additional parameter, $\Lambda$, which we may use to our advantage to obtain meaningful convergence results even when $D$ scales polynomially. Indeed, we observe that each occurrence of $\tilde{C}$ in our construction is normalized by at least $\Lambda^2$. Thus, since $\tilde{C} = O(D^2)$, by replacing $\Lambda \leftarrow \Lambda^{2u+1}$ in our analysis, and assuming $\Lambda$ is polynomial in t, we can obtain (ignoring polylog factors) $\tilde{O}\left(\frac{1}{t^\gamma}\right)$ convergence of the final iterate of SGD, for any $\gamma < 1$. Note that this change simply modifies the definition of $r_t$ by a constant factor. Thus, our convergence guarantee continues to hold with minor modifications to the choice of constants in our analysis.*

A direct consequence of Theorem 5 and the fact that $\|\boldsymbol{w}_0 - \boldsymbol{w}^*_{h+1,2i}\|_2^2 \leq 2\|\boldsymbol{w}_0 - \boldsymbol{w}_{h,i}\|_2^2 + 2\|\boldsymbol{w}^*_{h,i} - \boldsymbol{w}_{h+1,2i}\|_2^2 \leq 4\nu_1 \rho^h$ by Theorem 4 is the following Corollary, which guides our SGD budget allocation strategy.

**Corollary 3.** *Consider a tree node $(h, i)$ with mixture weights $\boldsymbol{\alpha}_{h,i}$ and optimal learning parameter $\boldsymbol{w}^*_{h,i}$. Assuming we start at a initial point $\boldsymbol{w}_0$ such that $\left\|\boldsymbol{w}_0 - \boldsymbol{w}^*_{h,i}\right\|_2^2 \leq \nu_1 \rho^h$ and take $t = \lambda(h+1)$ SGD steps using the child node distribution $p^{(\boldsymbol{\alpha}^*_{h+1,2i})}$ where, $\lambda(h+1)$ is chosen to satisfy*

$$\frac{G(4\nu_1\rho^h, \mathcal{G}_*)}{\lambda(h+1)+E} + \frac{8\lambda(h+1)\tilde{C}(D^2, D\sqrt{\mathcal{G}_*})}{\mu(\lambda(h+1)+E)\Lambda^7} + \frac{4\sqrt{2\log(\Lambda^8)}\sqrt{\hat{C}(k)}}{\mu(\lambda(h+1)+E)^{\alpha_{k+1}}} \leq \nu_1\rho^{h+1}, \tag{13}$$

*then by Theorem 5, with probability at least $1 - \frac{1}{\Lambda^7}$ we have $\|\boldsymbol{w}_t - \boldsymbol{w}^*_{h+1,2i}\|_2^2 \leq \nu_1\rho^{h+1}$.*

*In particular, if we assume that $D^2 = K(t)d_0^2$ for some $K(t)$ such that $K(t)/\Lambda^6 = \hat{K} = O(1)$ (refer to Remark 3 for why this particular assumption is reasonable), then when $G = Ed_0^2$ (i.e. $Ed_0^2 \geq \frac{8\mathcal{G}_*}{\mu}$)*

and $\hat{C}(0) = \frac{8d_0^2(4\beta^2 d_0^2 + \mathcal{G}_*)}{1+E}$ *(note that a similar statement can be made if the third term inside the* max *in* $\check{C}$ *from Theorem 5, instead of the first term, is maximal), taking* $k = 0$, *we may choose* $\lambda(h)$ ***independently of*** $h$:

$$
\lambda(h+1) = \lambda = \left( \frac{1}{\rho\sqrt{1+E}} \left( 4E + 64\sqrt{2}\kappa\hat{K} + \frac{16\sqrt{E\hat{K}}}{\sqrt{\mu}\Lambda^3} \right. \right.
$$
$$
\left. \left. + 128\kappa\sqrt{\log(\Lambda^8)} + \frac{16\sqrt{2E\log(\Lambda^8)}}{\sqrt{\mu}} \right) \right)^2 - E. \qquad (14)
$$

We will proceed in bounding the final iterate of SGD as follows:

- One main difficulty in analyzing the final iterate of SGD in our setting is our relaxed assumption on the norm of the gradient – namely, we assume that the norm of the gradient is bounded *only* at the optimal $\boldsymbol{w}^*$. We thus will rely on Lemma 2 and Lemma 3 to proceed with our analysis.
- In Lemma 4 and Lemma 5, we will derive a bound on the distance from the optimal solution which takes a form similar to that in the expected value analysis of [27, 7].
- Afterwards, we will define a sequence of random variables $r_t$ and $V_t$, in order to prove a high-probability result for $d_t^2 > r_t$ in Lemma 8.
- Given this high probability result, it is then sufficient to obtain an almost sure bound on $r_t$. We will proceed with bounding this quantity in several stages:
  - First, we obtain a useful bound on $r_t$ in Lemma 9 which normalizes the global diameter term $D$ by a term which is polynomial in our tunable parameter $\Lambda$. Note that this step is crucial to our analysis, as $D$ can potentially grow *polynomially* in the number of SGD steps $T$ under our assumptions, as we note in Remark 3.
  - Given this bound, we are left only to bound the $V_t$ term. We first obtain a crude bound on this term in Lemma 10, which would allow us to achieve a $\tilde{O}(1/\sqrt{t})$ converge guarantee. We then refine this bound in Corollary 4, which allows us to give a convergence guarantee of $\tilde{O}(K(\gamma)/t^\gamma)$ for any $\gamma < 1$ and for some constant $K(\gamma)$. We discuss how this refinement affects constant and $\log \Lambda$ factors in our convergence guarantee in Remark 4.
  - Finally, we collect our results to obtain our final bound on $r_{t+1}$ in Corollary 5.
- With a bound on $r_{t+1}$ and a high probability guarantee of $d_{t+1}$ exceeding $r_{t+1}$, we can finally obtain our high probability guarantee on error the final SGD iterate in Theorem 5.

Since quite a lot of notation will be introduced in this section, we provide a summary of parameters used here:

| Parameter | Value | Description |
|---|---|---|
| $g(\boldsymbol{w}_t; z_t),\ g_t$ | $\nabla f(\boldsymbol{w}_t; z_t)$ | Interchangeable notation for stochastic gradient |
| $\kappa$ | $\frac{\beta}{\mu}$ | The condition number |
| $d_t$ | $\|\boldsymbol{w}_t - \boldsymbol{w}^*\|_2^2$ | The distance of the $t$th iterate of SGD |
| $\eta_t$ | $\frac{2}{\mu(t+E)}$ | The step size of SGD |
| $E$ | $4096\kappa^2 \log \Lambda^4$ | |
| $T$ | | The number of SGD iterations |
| $\Lambda$ | $\geq T + 1$ | Tunable parameter to control high probability bound |
| $M_t$ | $\langle \nabla F(\boldsymbol{w}_t) - g_t, \boldsymbol{w}_t - \boldsymbol{w}^* \rangle$ | |
| $\varrho_t$ | $2d_t\sqrt{8\beta^2 d_t^2 + 2\mathcal{G}_*}$ | Upper bound on the martingale difference sequence |
| $D$ | $\sup_{t=0,\ldots,T} d_t$ | The uniform diameter bound (discussed in Remark 3) |

We begin by noting that crucial to our analysis is deriving bounds on our stochastic gradient, since we assume the norm of the stochastic gradient is bounded *only* at the origin. The following results are the versions of Lemma 2 from [27] restated as almost sure bounds.

**Lemma 2** (Sample path version of Lemma 2 from [27]). *Under Assumption 1 and Assumption 2, the following bound on the norm of the stochastic gradient holds almost surely.*

$$\|g(\boldsymbol{w}_t, Z_t)\|^2 \le 4\beta\kappa(F(\boldsymbol{w}_t) - F(\boldsymbol{w}^*)) + 2\mathcal{G}_* \tag{15}$$

*Proof.* As in [27], we note that since

$$\|a - b\|^2 \ge \frac{1}{2}\|a\|^2 - \|b\|^2, \tag{16}$$

we may obtain the following bound:

$$
\begin{aligned}
\frac{1}{2}\|\nabla f(\boldsymbol{w}_t; z)\|^2 - \|\nabla f(\boldsymbol{w}^*; z)\|^2 &\le \|\nabla f(\boldsymbol{w}_t; z) - \nabla f(\boldsymbol{w}^*; z)\|^2 \\
&\le \beta^2\|\boldsymbol{w}_t - \boldsymbol{w}^*\|^2 \qquad \text{by } \beta\text{-smoothness of } f \\
&\le \frac{2\beta^2}{\mu}(F(\boldsymbol{w}_t) - F(\boldsymbol{w}^*)) \qquad \text{by } \mu\text{-strong convexity of } F
\end{aligned}
$$

Rearranging, we have that

$$\|\nabla f(\boldsymbol{w}_t; z)\|^2 \le 4\beta\kappa(F(\boldsymbol{w}_t) - F(\boldsymbol{w}^*)) + 2\mathcal{G}_*, \tag{17}$$

as desired. $\qquad\square$

**Lemma 3** (Centered sample path version of Lemma 2 from [27]). *Under Assumption 1 and Assumption 2, for any random realization of z, the following bound holds almost surely:*

$$\|\nabla f(\boldsymbol{w}_t; z) - \nabla F(\boldsymbol{w}_t)\|^2 \le 8\beta^2\|\boldsymbol{w}_t - \boldsymbol{w}^*\|^2 + 2\mathcal{G}_* \tag{18}$$

*Proof.* The proof proceeds similarly to Lemma 2, replacing the stochastic gradient with the mean-centered version to obtain:

$$
\begin{aligned}
&\frac{1}{2}\|\nabla f(\boldsymbol{w}_t; z) - \mathbb{E}[\nabla f(\boldsymbol{w}_t; z)]\|^2 - \|\nabla f(\boldsymbol{w}^*; z) - \mathbb{E}[\nabla f(\boldsymbol{w}^*; z)]\|^2 \\
&\le \|\nabla f(\boldsymbol{w}_t; z) - \nabla f(\boldsymbol{w}^*; z) - \mathbb{E}[\nabla f(\boldsymbol{w}_t; z)] + \mathbb{E}[\nabla f(\boldsymbol{w}^*; z)]\|^2 \\
&\le 2(\|\nabla f(\boldsymbol{w}_t; z) - \nabla f(\boldsymbol{w}^*; z)\|^2 + \|\mathbb{E}[\nabla f(\boldsymbol{w}_t; z)] - \mathbb{E}[\nabla f(\boldsymbol{w}^*; z)]\|^2) \\
&\le 2(\|\nabla f(\boldsymbol{w}_t; z) - \nabla f(\boldsymbol{w}^*; z)\|^2 + \mathbb{E}[\|\nabla f(\boldsymbol{w}_t; z) - \nabla f(\boldsymbol{w}^*; z)]\|^2]) \\
&\le 4\beta^2\|\boldsymbol{w}_t - \boldsymbol{w}^*\|^2
\end{aligned}
$$

Now, rearranging terms, and recalling that $\mathbb{E}[\nabla f(\boldsymbol{w}^*; z)] = \nabla F(\boldsymbol{w}^*) = 0$, we have

$$
\begin{aligned}
\|\nabla f(\boldsymbol{w}_t; z) - \nabla F(\boldsymbol{w}_t)\|^2 &= \|\nabla f(\boldsymbol{w}_t; z) - \mathbb{E}[\nabla f(\boldsymbol{w}_t; z)]\|^2 \\
&\le 8\beta^2\|\boldsymbol{w}_t - \boldsymbol{w}^*\|^2 + 2\|\nabla f(\boldsymbol{w}^*; z)\|^2 \\
&\le 8\beta^2\|\boldsymbol{w}_t - \boldsymbol{w}^*\|^2 + 2\mathcal{G}_*
\end{aligned}
$$

as desired. $\qquad\square$

Given these bounds on the norm of the stochastic gradient, we are now prepared to begin deriving high probability bounds on the optimization error of the final iterate.

**Lemma 4.** *Suppose $F$ and $f$ satisfy Assumption 1 and Assumption 2. Consider the stochastic gradient iteration $\boldsymbol{w}_{t+1} = \boldsymbol{w}_t - \eta_t\nabla f(\boldsymbol{w}_t; z_t)$, where $z$ is sampled randomly from a distribution $p(z)$. Let $\boldsymbol{w}^* = \arg\min_{\boldsymbol{w}} F(\boldsymbol{w})$. Let $M_t = \langle\nabla F(\boldsymbol{w}_t) - g(\boldsymbol{w}_t, Z_t), \boldsymbol{w}_t - \boldsymbol{w}^*\rangle$, where $g(\boldsymbol{w}, z) = \nabla f(\boldsymbol{w}, z)$. Additionally, let us adopt the notation $d_t = \|\boldsymbol{w}_t - \boldsymbol{w}^*\|_2$. Then the iterates satisfy the following inequality:*

$$d_{t+1}^2 \le (1 - \mu\eta_t)d_t^2 + 2\mathcal{G}_*\eta_t^2 + 2\eta_t M_t \tag{19}$$

*as long as $0 < \eta_t \le \frac{1}{2\beta\kappa}$, where $\kappa = \frac{\beta}{\mu}$.*

*Proof.* The proof crucially relies on techniques employed in [27], and in particular, on Lemma 2, We now apply this result to bound $d_{t+1}$ :

$$
\begin{aligned}
\|\boldsymbol{w}_{t+1} - \boldsymbol{w}^*\|^2 &= \|\boldsymbol{w}_t - \eta_t g(\boldsymbol{w}_t; z_t) - \boldsymbol{w}^*\|^2 && \text{by definition of SGD} \\
&= \|\boldsymbol{w}_t - \boldsymbol{w}^*\|^2 + \eta_t^2 \|g(\boldsymbol{w}_t; z_t)\|^2 \\
&\quad - 2\eta_t \langle g(\boldsymbol{w}_t; z_t), \boldsymbol{w}_t - \boldsymbol{w}^* \rangle \\
&\leq \|\boldsymbol{w}_t - \boldsymbol{w}^*\|^2 + 2\eta_t^2 (\mathcal{G}_* + 2\beta\kappa(F(\boldsymbol{w}_t) - F(\boldsymbol{w}^*))) \\
&\quad - 2\eta_t(\langle \nabla F(\boldsymbol{w}_t), \boldsymbol{w}_t - \boldsymbol{w}^* \rangle \\
&\qquad + \langle g_t - \nabla F(\boldsymbol{w}_t), \boldsymbol{w}_t - \boldsymbol{w}^* \rangle) && \text{by Lemma 2} \\
&\leq \|\boldsymbol{w}_t - \boldsymbol{w}^*\|^2 + 2\eta_t^2 (\mathcal{G}_* + 2\beta\kappa(F(\boldsymbol{w}_t) - F(\boldsymbol{w}^*))) \\
&\quad - 2\eta_t(F(\boldsymbol{w}_t) - F(\boldsymbol{w}^*) + \frac{\mu}{2}\|\boldsymbol{w}_t - \boldsymbol{w}^*\|^2 \\
&\qquad + \langle g_t - \nabla F(\boldsymbol{w}_t), \boldsymbol{w}_t - \boldsymbol{w}^* \rangle) && \text{by } \mu\text{-s.c. of } F \\
&= (1 - \mu\eta_t)\|\boldsymbol{w}_t - \boldsymbol{w}^*\|^2 \\
&\quad - 2\eta_t(1 - 2\beta\kappa\eta_t)(F(\boldsymbol{w}_t) - F(\boldsymbol{w}^*)) \\
&\quad - 2\eta_t\langle g_t - \nabla F(\boldsymbol{w}_t), \boldsymbol{w}_t - \boldsymbol{w}^* \rangle + 2\mathcal{G}_*\eta_t^2 \\
&\leq (1 - \mu\eta_t)d_t^2 + 2\mathcal{G}_*\eta_t^2 + 2\eta_t M_t && \text{assuming } \eta_t \leq \frac{1}{2\beta\kappa}
\end{aligned}
$$

which is the desired result. $\qquad\square$

Now given this recursion, we may derive a bound on $d_{t+1}$ in a similar form as expected value results from Theorem 2 from [27] and Theorem 4.7 in [7]. Namely,

**Lemma 5.** *Using the same assumptions and notation as in Lemma 4, by choosing* $\eta_t = \frac{2}{\mu(t+E)}$, *where* $E \geq 4\kappa^2$ *we have the following bound on the distance from the optimum:*

$$
\begin{aligned}
d_t^2 &\leq \frac{G(d_0^2, \mathcal{G}_*)}{t + E} + \sum_{i=0}^{t-1} c(i, t-1) M_i \\
&\leq \frac{G(d_0^2, \mathcal{G}_*)}{t + E} + \frac{4}{\mu(t+E)} \sum_{i=0}^{t} M_i
\end{aligned}
$$

*where*

$$
G(d_0^2, \mathcal{G}_*) = \max\{E d_0^2, \frac{8\mathcal{G}_*}{\mu^2}\}, \text{ and } c(i, t) = 2\eta_i \prod_{j=i+1}^{t} (1 - \mu\eta_j)
$$

*Proof.* We first note that our choice of $\eta_t$ does indeed satisfy $\eta_t \leq \frac{1}{2\beta\kappa}$, so we may apply Lemma 4. As in the aforementioned theorems, our proof will proceed inductively.

Note that the base case of $t = 0$ holds trivially by construction. Now let us suppose the bound holds for some $l < t$. Then, using the recursion derived in Lemma 4, we have that

$$
\begin{aligned}
d_{l+1}^2 &\le (1 - \mu\eta_l)d_l^2 + 2\mathcal{G}_*\eta_t^2 + 2\eta_t M_t \\
&\le (1 - \mu\eta_l)\left(\frac{G(d_0^2, \mathcal{G}_*)}{l + E} + \sum_{i=0}^{l-1} c(i, l-1)M_i\right) + 2\mathcal{G}_*\eta_l^2 + 2\eta_l M_l \\
&= (1 - \mu\eta_l)\frac{G(d_0^2, \mathcal{G}_*)}{l + E} + 2\mathcal{G}_*\eta_l^2 + \sum_{i=0}^{l} c(i, l)M_i \\
&= G(d_0^2, \mathcal{G}_*)\frac{l + E - 2}{(l + E)^2} + \frac{8\mathcal{G}_*}{\mu^2(l + E)^2} + \sum_{i=0}^{l} c(i, l)M_i \\
&= G(d_0^2, \mathcal{G}_*)\frac{l + E - 1}{(l + E)^2} - \frac{G(d_0^2, \mathcal{G}_*)}{(l + E)^2} + \frac{8\mathcal{G}_*}{\mu^2(l + E)^2} + \sum_{i=0}^{l} c(i, l)M_i
\end{aligned}
$$

Now note that, by definition of $G(d_0^2, \mathcal{G}_*)$, we have that

$$
-\frac{G(d_0^2, \mathcal{G}_*)}{(l + E)^2} + \frac{8\mathcal{G}_*}{\mu^2(l + E)^2} \le 0 \tag{20}
$$

Therefore, we find that

$$
\begin{aligned}
d_{l+1}^2 &\le G(d_0^2, \mathcal{G}_*)\frac{l + E - 1}{(l + E)^2} + \sum_{i=0}^{l} c(i, l)M_i \\
&= G(d_0^2, \mathcal{G}_*)\frac{(l + E)^2 - 1}{(l + E)^2}\frac{1}{t + E + 1} + \sum_{i=0}^{l} c(i, l)M_i \\
&\le \frac{G(d_0^2, \mathcal{G}_*)}{(l + 1) + E} + \sum_{i=0}^{l} c(i, l)M_i
\end{aligned}
$$

Thus, the result holds for all $t$.

We now note that $c(i, t) \le \frac{4}{\mu(t+E)}$. Observe that

$$
\begin{aligned}
c(i, t) &= 2\eta_i \prod_{j=i+1}^{t} (1 - \mu\eta_j) \\
&= \frac{4}{\mu(i + E)} \prod_{j=i+1}^{t} \frac{j + E - 2}{j + E} \\
&= \frac{4}{\mu(i + E)} \frac{i + E - 1}{t + E} \\
&\le \frac{4}{\mu(t + E)}
\end{aligned}
$$

$\square$

Now, in order to obtain a high probability bound on the final iterate of SGD, we need to obtain a concentration result for $\sum_{i=0}^{t} M_i$. We note that, from Lemma 3, we obtain an upper bound on the magnitude of $M_i$ :

$$
\begin{aligned}
|M_t| &\le \|g(\boldsymbol{w}_t; z_t) - \nabla F(\boldsymbol{w}_t)\|\|\boldsymbol{w}_t - \boldsymbol{w}^*\| \\
&\le \sqrt{8\beta^2 d_t^2 + 2\mathcal{G}_* d_t}.
\end{aligned}
$$

We consider the usual filtration $\mathcal{F}_t$ that is generated by $\{z_i\}_{i \leq t}$ and $\mathbf{w}_0$. Just for completeness of notation we set $z_0 = 0$ (no gradient at step 0).

By this construction, we observe that $M_t$ is a martingale difference sequence with respect to the filtration $\mathcal{F}_t$. In other words, $S_t = \sum_{s=1}^{t} M_s$ is a martingale.

**Lemma 6.** $\mathbb{E}[M_t \mid \mathcal{F}_{t-1}] = 0, \ \forall t > 0.$

*Proof.* Given the filtration, $\mathcal{F}_{t-1}$, $\mathbf{w}_0, z_1 \ldots z_{t-1}$ is fixed. This implies that $\mathbf{w}_t$ is fixed. However, conditioned on $\{z_i\}_{i<t}$, $z_t$ is randomly sampled from $p(z)$. Therefore, $\mathbb{E}[g(\mathbf{w}_t, z_t) - \nabla F(\mathbf{w}_t) \mid \mathcal{F}_{t-1}] = \mathbb{E}_{z_t \mid \mathcal{F}_{t-1}}[g(\mathbf{w}_t, z_t) - \nabla F(\mathbf{w}_t) \mid \mathbf{w}_t] = \mathbb{E}_{z_t \sim p(z)}[g(\mathbf{w}_t, z_t) - \nabla F(\mathbf{w}_t) \mid \mathbf{w}_t] = 0$. Hence, $\mathbb{E}[M_t \mid \mathcal{F}_{t-1}] = 0$ □

Recall that, $M_s$ is uniformly upper bounded by $\varrho_t = d_t \sqrt{8\beta^2 d_t^2 + 2\mathcal{G}_*}$. Thus, we have that $\varrho_t^2 \leq d_t^2 (8\beta^2 d_t^2 + 2\mathcal{G}_*)$.

Let $D = \sup_{0 \leq t \leq T} d_t$. Then, $|M_t| \leq d_t \sqrt{8\beta^2 d_t^2 + 2\mathcal{G}_*} \leq \tilde{C}(D^2, D\sqrt{\mathcal{G}_*}) = D\sqrt{8\beta^2 D^2 + 2\mathcal{G}_*}$.

In order to obtain a high probability bound on the final SGD iterate, we will introduce the following sequence of random variables and events, and additionally constants $c'(t)$ to be decided later.

1. **Initialization at** $t = 0$**:** Let $V_0 = \frac{8 d_0^2 (4\beta^2 d_0^2 + \mathcal{G}_*)}{1+E}$, $r_0 = d_0^2$, and take $\mathcal{A}_0$ to be an event that is true with probability 1. Let $M_0 = 0$. $\Pr(\mathcal{E}_0)=1$, $\delta_0 = 0$.

2. $r_t = \frac{G}{t+E} + \frac{4}{\mu(t+E)}(t-1)\delta_{t-1}\tilde{C}(D^2, D\sqrt{\mathcal{G}_*}) + \frac{4}{\mu}\sqrt{\frac{2\log(\Lambda^8/c'(t))}{t+E}}\sqrt{V_{t-1}}$

3. $V_t = \frac{1}{t+E+1} \sum_{i=0}^{t} 8r_i(4\beta^2 r_i + \mathcal{G}_*)$.

4. Event $\mathcal{A}_t$ is all sample paths satisfying the condition: $d_t^2 \leq r_t$.

5. Let $\mathcal{E}_t = \bigcap_{i \leq t} \mathcal{A}_i$. Further, let $\Pr(\mathcal{E}_t^c)/\Pr(\mathcal{E}_t) = \delta_t$.

We now state a conditional form of the classic Azuma-Hoeffding inequality that has been tailored to our setting, and provide a proof for completeness.

**Lemma 7** (Azuma-Hoeffding with conditioning)**.** *Let $S_n = f(z_1 \ldots z_n)$ be a martingale sequence with respect to the filtration $\mathcal{F}_n$ generated by $z_1 \ldots z_n$. Let $\psi_n = S_n - S_{n-1}$. Suppose $|\psi_n| \leq c_n(z_1 \ldots z_{n-1})$ almost surely. Suppose $E[\psi_n \mid \mathcal{F}_{n-1}] = 0$.*

*Let $\mathcal{A}_{n-1}$ be the event that $c_n \leq d_n$, where $\mathcal{A}_{n-1}$ is defined on the filtration $\mathcal{F}_{n-1}$, and $\underline{d}_n$ is a constant dependent only on the index $n$. Define $\mathcal{E}_n = \bigcap_{i \leq n} \mathcal{A}_i$. Further suppose that that $\exists \bar{R}$ large enough such that $|\psi_n| \leq \bar{R}$ almost surely. Finally let $\Pr(\mathcal{E}_n^c)/\Pr(\mathcal{E}_n) = \delta_n$. Then,*

$$\Pr\left(S_n \geq \gamma + n\delta_n\bar{R} \mid \mathcal{E}_n\right) \leq \exp\left(-\frac{\gamma^2}{2\sum_{i=1}^{n} d_i^2}\right) \tag{21}$$

*Proof.* We first observe that $\mathbb{E}[\psi_i \mid \mathcal{F}_{i-1}] = 0$. Therefore, for $i \leq n$ we have:

$$\begin{aligned}
|\mathbb{E}[\psi_i \mid \mathcal{E}_n, \mathcal{F}_{i-1}]| &= \frac{\Pr(\mathcal{E}_n^c)}{\Pr(\mathcal{E}_n)}|\mathbb{E}[\psi_i \mid \mathcal{E}_n^c, \mathcal{F}_{i-1}]| \\
&\leq \frac{\Pr(\mathcal{E}_n^c)}{\Pr(\mathcal{E}_n)}\bar{R} \\
&\leq \delta_n \bar{R} \tag{22}
\end{aligned}$$

Consider the sequence $S_i' = S_i - \sum_{j=1}^{i} \mathbb{E}[\psi_j \mid \mathcal{F}_{j-1}, \mathcal{E}_n]$ for $i \leq n$.

$$\begin{aligned}
\Pr(S_i' \geq \gamma \mid \mathcal{E}_n) &\leq e^{-\theta\gamma}\mathbb{E}[e^{\theta S_i'} \mid \mathcal{E}_n] \\
&= e^{-\theta\gamma}\mathbb{E}[\mathbb{E}[e^{\theta S_i'} \mid \mathcal{E}_n, \mathcal{F}_{i-1}] \mid \mathcal{E}_n] \\
&= e^{-\theta\gamma}\mathbb{E}[e^{\theta S_{i-1}'}\mathbb{E}[e^{\theta(\psi_i - \mathbb{E}[\psi_i \mid \mathcal{F}_{i-1}, \mathcal{E}_n])} \mid \mathcal{E}_n, \mathcal{F}_{i-1}] \mid \mathcal{E}_n]
\end{aligned}$$

$$\tag{23}$$

Observe that $\mathbb{E}[\psi_i - \mathbb{E}[\psi_i \mid \mathcal{F}_{i-1}, \mathcal{E}_n] \mid \mathcal{F}_{i-1}, \mathcal{E}_n] = 0$. i.e. $\psi_i - \mathbb{E}[\psi_i \mid \mathcal{F}_{i-1}, \mathcal{E}_n]$ is a mean 0 random variable with respect to the conditioning events $\mathcal{F}_{i-1}, \mathcal{E}_n$.

Further, for any sample path where $\mathcal{E}_n$ holds, we almost surely have $|\psi_i - \mathbb{E}[\psi_i \mid \mathcal{F}_{i-1}, \mathcal{E}_n]| \le 2c_i(z_1, z_2 \ldots z_{i-1}) \le 2d_i$.

Therefore, $\mathbb{E}[e^{\theta(\psi_i - \mathbb{E}[\psi_i \mid \mathcal{F}_{i-1}, \mathcal{E}_n])} \mid \mathcal{E}_n, \mathcal{F}_{i-1}] \le e^{4\theta d_i^2/2}$

Therefore, Equation 23 yields the following:

$$\Pr(S_i' \ge \gamma \mid \mathcal{E}_n) \le e^{-\theta\gamma}\mathbb{E}[e^{\theta S_{i-1}'} \mid \mathcal{E}_n][e^{\frac{4\theta d_i^2}{2}}]$$
$$= e^{-\theta\gamma}e^{\theta \sum_{j=1}^{i} 4d_j^2/2} \tag{24}$$

Let $\theta = \frac{\gamma}{\sum_{i=1}^{n} 4d_i^2}$. Then, we have for $i = n$:

$$\Pr\left(S_n \ge \gamma + \sum_{i=1}^{n} \mathbb{E}[\psi_i \mid \mathcal{F}_{i-1}, \mathcal{E}_i] \,\middle|\, \mathcal{E}_n\right) \le \exp\left(-\frac{\gamma^2}{8\sum_{i=1}^{n} d_i^2}\right)$$
$$\overset{a}{\Rightarrow}\Pr\left(S_n \ge \gamma + n\delta_n\bar{R} \mid \mathcal{E}_n\right) \le \exp\left(-\frac{\gamma^2}{8\sum_{i=1}^{n} d_i^2}\right) \tag{25}$$

(a) - This is obtained by substituting the almost sure bound Equation 22 for all $i \le n$.

$\square$

Using our iterative construction and the conditional Azuma-Hoeffding inequality, we obtain the following high probability bound:

**Lemma 8.** *Under the construction specified above, we have the following:*

$$\Pr(d_{t+1}^2 > r_{t+1} \mid \mathcal{E}_t) \le \frac{c'(t+1)}{\Lambda^8} \tag{26}$$

*When $c'(i) = 1$, we have:*

$$\Pr(\mathcal{E}_{t+1}^c) \le \frac{t+1}{\Lambda^8} \tag{27}$$

*Proof.* By the conditional Azuma-Hoeffding Inequality (Lemma 7), we have the following chain:

$$\Pr(\mathcal{A}_{t+1}^c \mid \mathcal{A}_i, \ i \le t) = \Pr(d_{t+1}^2 > r_{t+1} \mid \mathcal{A}_i, \ i \le t)$$
$$\le \Pr\left(\frac{4}{\mu(t+1+E)} \sum_{i=1}^{t}(M_i - \delta_t\tilde{C}(D^2, D\sqrt{\mathcal{G}_*})) > \right.$$
$$\left. \frac{4}{\mu(t+1+E)}\sqrt{\sum_{i=0}^{t}\varrho_i^2}\sqrt{2\log\left(\frac{\Lambda^8}{c'(t+1)}\right)} \,\middle|\, \mathcal{A}_i, \ i \le t\right)$$
$$\overset{a}{\le} \exp\left(-\frac{(2\log(\frac{\Lambda^8}{c'(t+1)}))\sum_{i=0}^{t}\varrho_i^2}{2\sum_{i=0}^{t}\varrho_i^2}\right)$$
$$= \frac{c'(t+1)}{\Lambda^8}$$

(a)- We set $\psi_i$ in Lemma 7 to be the variables $M_i$, filtrations $\mathcal{F}_t$ to be that generated by $z_t \sim p(z)$ (and $\mathbf{w}_0$) in the stochastic gradient descent steps. $c_t$ (in Lemma 7) set to $\varrho_t$, $d_t$ (in Lemma 7) is set to $r_t$, $\bar{R}$ (in Lemma 7) is set to $\tilde{C}(D^2, D\sqrt{\mathcal{G}_*})$ and $\delta_t$ (in Lemma 7) is set to $\Pr(\mathcal{E}_t^c)/\Pr(\mathcal{E}_t)$. Now, if we apply Lemma Lemma 7 to the sequence $M_i$ with the deviation $\gamma$ set to $\sqrt{\sum_{i=0}^{t}\varrho_i^2}\sqrt{2\log\left(\frac{\Lambda^8}{c'(t+1)}\right)}$, we obtain the inequality.

$$\Pr(\mathcal{E}_{t+1}^c) \le \sum_{i=1}^{t+1} \Pr(\min\{j : d_j^2 > r_j\} = i)$$

$$\le \sum_{i=1}^{t+1} \Pr(\mathcal{A}_i^c \mid \mathcal{A}_j, \ j < i) = \sum_{i=1}^{t+1} \frac{c'(i)}{\Lambda^8} \tag{28}$$

Choosing $c'(i) = 1$, we thus obtain our desired result.

$\square$

From Lemma 8, we have a high probability bound on the event that $d_t^2 > r_t$. In order to translate this to a meaningful SGD convergence result, we will have to substitute for $\delta_t$. We thus upper bound $r_t$ as follows:

**Lemma 9.** *Under the above construction, where $c'(i)$ is chosen to be 1, we have the following almost sure upper bound on $r_t$, $\forall t \le \Lambda$*

$$r_t \le \frac{G(d_0^2, \mathcal{G}_*)}{t + E} + \frac{8t\tilde{C}(D^2, D\sqrt{\mathcal{G}_*})}{\mu(t+E)\Lambda^7} + \frac{4\sqrt{2\log(\Lambda^8)}\sqrt{V_{t-1}}}{\mu\sqrt{t+E}} \tag{29}$$

*where $\tilde{C}(D^2, D\sqrt{\mathcal{G}_*}) = D\sqrt{8\beta^2 D^2 + 2\mathcal{G}_*}$, and $D$ is taken to be a uniform diameter bound*[3].

*Proof.* From Lemma 8, we have: $\delta_t = \frac{\Pr(\mathcal{E}_t^c)}{\Pr(\mathcal{E}_t)} \le \frac{t}{\Lambda^8 - t} \le \frac{2}{\Lambda^7}$. Here, we assume that $\Lambda > 2$. Substituting in the expression for $r_t$, we have the result. $\square$

Given this bound from Lemma 9, we now must construct an upper bound on $V_t$. We will proceed in two steps, first deriving a crude bound on $V_t$, and then by iteratively refining this bound. We now derive the crude bound.

**Lemma 10.** *The following bound on $V_t$ holds almost surely:*

$$V_t \le \check{C}\log\Lambda^8 \tag{30}$$

*assuming that we choose*

$$E \ge 128\beta^2 c_2^2 \log\Lambda^8$$

$$\check{C} \ge \max\left\{ \frac{V_0}{\log\Lambda^8}, (8\mathcal{G}_* c_2 + \min\{2/E, 1\})^2, \left(\frac{64\beta^2 c_1^2}{(1+E)\log\Lambda^8} + \frac{8\mathcal{G}_* c_1}{\log\Lambda^8}\right)^2 \right\}$$

$$c_1 = G(d_0^2, \mathcal{G}_*) + \frac{8\tilde{C}(D^2, D\sqrt{\mathcal{G}_*})}{\mu\Lambda^6}$$

$$c_2 = \frac{4\sqrt{2}}{\mu}$$

$$\Lambda \ge t + 1$$

*Proof.* We will prove the claim inductively.

We note that the base case when $t = 0$ holds by construction, assuming that $\check{C} \ge \frac{V_0}{\log\Lambda^4}$.

Now let us suppose that our claim holds until some $t$. Then by applying the bound on $r_t$ derived in Lemma 9, we have the following bound:

$$r_{t+1} \le \frac{G(d_0^2, \mathcal{G}_*)}{t+1+E} + \frac{8(t+1)\tilde{C}(D^2, D\sqrt{\mathcal{G}_*})}{\mu(t+1+E)\Lambda^7} + \frac{4\sqrt{2\check{C}}\log\Lambda^8}{\mu\sqrt{t+E+1}}$$

$$\le \frac{c_1}{t+E+1} + \frac{c_2\sqrt{\check{C}}\log\Lambda^8}{\sqrt{t+E+1}},$$

where $c_1 = G(d_0^2, \mathcal{G}_*) + \frac{8\check{C}(D^2, D\sqrt{\mathcal{G}_*})}{\mu\Lambda^6}$ and $c_2 = \frac{4\sqrt{2}}{\mu}$. Plugging in this bound to our definition of $V_{t+1}$, we obtain:

$$V_{t+1} = \frac{t+1+E}{t+2+E}V_t + 32\beta^2 r_{t+1}^2 + 8\mathcal{G}_* r_{t+1}$$

$$\leq \frac{\check{C}\log\Lambda^8}{t+E+2}\left((t+E+1) + \frac{64\beta^2 c_2^2 \log\Lambda^8}{t+E+1} + \frac{8\mathcal{G}_* c_2}{\sqrt{\check{C}(t+E+1)}}\right)$$

$$+ \frac{1}{(t+E+1)(t+E+2)}\left(\frac{64\beta^2 c_1^2}{(t+E+1)} + 8\mathcal{G}_* c_1\right)$$

$$\overset{\text{shown below}}{\leq} \check{C}\log\Lambda^8$$

Rearranging, we find that we equivalently need:

$$\frac{64\beta^2 c_1^2}{(t+E+1)\log\Lambda^8} + \frac{8\mathcal{G}_* c_1}{\log\Lambda^8} \leq \sqrt{\check{C}}(\sqrt{\check{C}}(t+E+1 - 64\beta^2 c_2^2 \log\Lambda^8)$$

$$- 8\mathcal{G}_* c_2 \sqrt{t+E+1}).$$

Now, setting $E = 2 * 64\beta^2 c_2^2 \log\Lambda^8$, we find that a sufficient condition to complete our induction hypothesis is:

$$\frac{64\beta^2 c_1^2}{(t+E+1)\log\Lambda^8} + \frac{8\mathcal{G}_* c_1}{\log\Lambda^8} \leq \sqrt{\check{C}}((\sqrt{\check{C}} - 4\mathcal{G}_* c_2)(t+1) + (\sqrt{\check{C}} - 8\mathcal{G}_* c_2)E/2). \quad (31)$$

Now, observe that by choosing

$$\check{C} \geq \max\left\{(8\mathcal{G}_* c_2 + \min\{2/E, 1\})^2, \left(\frac{64\beta^2 c_1^2}{(1+E)\log\Lambda^8} + \frac{8\mathcal{G}_* c_1}{\log\Lambda^8}\right)^2\right\} \quad (32)$$

the sufficient condition Equation 31 is satisfied. Hence, our claim holds for all $t$. $\qquad\square$

Now given this crude upper bound, we may repeatedly apply Lemma 8 from [27] in order to obtain the following result:

**Corollary 4** (of Lemma 10 + Lemma 8 in [27]). *After $k \geq 0$ applications of Lemma 8 from [27], under the same assumptions as in Lemma 10, we have the following bound on $V_t$:*

$$V_t \leq \frac{\hat{C}(k)}{(t+E+1)^{\alpha_k}} \quad (33)$$

*where*

$$\hat{C}(k+1) = 2^{k+1}C(k+1) + V_0\frac{1+E}{(2+E)^{1-\alpha_{k+1}}}$$

$$C(k+1) = \frac{64\beta^2 c_1^2}{(E+1)^{2-\alpha_{k+1}}} + \frac{64\hat{C}(k)c_2^2}{\mu^2(E+1)^{\alpha_{k+1}}}$$

$$+ \frac{8\mathcal{G}_* c_1}{(E+1)^{1-\alpha_{k+1}}} + 8\frac{\mathcal{G}_*}{\mu}\sqrt{\hat{C}(k)}c_2$$

$$\alpha_{k+1} = \sum_{i=1}^{k+1}\frac{1}{2^i}$$

$$\hat{C}(0) = \check{C}\log\Lambda^8$$

$$\alpha_0 = 0$$

*where $E, \check{C}, c_1, c_2$ are defined as in Lemma 10.*

*Proof.* We will construct this bound inductively. We begin by noting that, when $k = 0$, the bound holds by Lemma 10. Now let us assume the bound holds until some $k$. Observe, then, that, by plugging into the bound in Lemma 9, we may write

$$V_{t+1} \leq \beta_t V_t + \gamma_t \tag{34}$$

where

$$\beta_t = \frac{t+1+E}{t+2+E}$$

$$\gamma_t = \frac{C(k+1)}{(t+E+1)^{\alpha_{k+1}}(t+E+2)}$$

$$C(k+1) = \frac{64\beta^2 c_1^2}{(E+1)^{2-\alpha_{k+1}}} + \frac{64\hat{C}(k)c_2^2}{\mu^2(E+1)^{\alpha_{k+1}}} + \frac{8\mathcal{G}_* c_1}{(E+1)^{1-\alpha_{k+1}}} + 8\frac{\mathcal{G}_*}{\mu}\sqrt{\hat{C}(k)}c_2$$

Now, we may apply Lemma 8 in [27] to obtain:

$$V_{t+1} \leq \sum_{i=0}^{t}\left(\prod_{j=i+1}^{t}\beta_j\right)\gamma_i + V_0\prod_{i=0}^{t}\beta_i$$

$$= \sum_{i=0}^{t}\frac{i+2+E}{t+2+E}\frac{C(k+1)}{(i+E+1)^{\alpha_{k+1}}(i+E+2)} + V_0\frac{1+E}{t+2+E}$$

$$\leq \frac{C(k+1)}{t+2+E}\int_{E}^{t+1+E}\frac{1}{x^{\alpha_{k+1}}}dx + V_0\frac{1+E}{t+2+E}$$

$$\leq \frac{C(k+1)}{(1-\alpha_k)(t+E+2)^{\alpha_{k+1}}} + V_0\frac{1+E}{t+E+2}$$

$$\leq \frac{\hat{C}(k+1)}{(t+E+2)^{\alpha_{k+1}}},$$

where $\hat{C}(k+1) = \frac{C(k+1)}{1-\alpha_{k+1}} + V_0\frac{1+E}{(2+E)^{1-\alpha_{k+1}}} = 2^{k+1}C(k+1) + V_0\frac{1+E}{(2+E)^{1-\alpha_{k+1}}}$.

Thus, our claim holds for all $k$. $\square$

**Remark 4.** *Note that while $\hat{C}(k)$ in Corollary 4 has complicated dependencies on $\mathcal{G}_*, d_0, \beta$, and $\mu$, it is straightforward to argue that $\hat{C}(k) \leq \rho_k \log \Lambda^8$ where $\rho_k$ is a constant that is independent of $\Lambda$. Indeed, note that, from Corollary 4, we have that*

$$\hat{C}(k+1) \leq 2^{k+1}\left[\frac{e_1}{(E+1)^{2-\alpha_{k+1}}} + \frac{e_2}{(E+1)^{1-\alpha_{k+1}}} + \frac{e_3}{(E+1)^{\alpha_{k+1}}}\hat{C}(k) + e_4\sqrt{\hat{C}(k)}\right]$$
$$+ \frac{e_5}{(2+E)^{1-\alpha_{k+1}}}$$

*for some $e_1, \ldots, e_5$ which are independent of $\Lambda$. Note that when $k = 0$, the claimed bound on $\hat{C}(0)$ holds by definition, for proper choice of $\rho_0$. Assuming the bound holds until $k$, we may construct a bound of the desired form by choosing $\rho_{k+1}$ as a function of the $e_i$s. Note that $E + 1 \geq 1$, and that each $e_i$ is independent of $\Lambda$, so $\rho_{k+1}$ is also independent of $\Lambda$. We may thus conclude that $\hat{C}(k) = O(\log \Lambda^8)$.*

We may collect these results to obtain:

**Corollary 5** (of Lemma 9 + Corollary 4). *Under the assumptions on $E$ in Lemma 10 and the definition of $\hat{C}(k)$ from Corollary 4, the following bound holds almost surely, for any $k \geq 0$,*

$$r_{t+1} \leq \frac{G(d_0^2, \mathcal{G}_*)}{t+E+1} + \frac{8(t+1)\tilde{C}(D^2, D\sqrt{\mathcal{G}_*})}{\mu(t+1+E)\Lambda^7} + \frac{4\sqrt{2\log(\Lambda^8)}\sqrt{\hat{C}(k)}}{\mu(t+E+1)^{\sum_{i=1}^{k+1}2^{-i}}} \tag{35}$$

We are now prepared to state and prove our main SGD result.

*Proof of Theorem 5.* The proof is an immediate consequence of Lemma 8 combined with Corollary 5. $\square$

# E Putting It Together: Tree-Search

## E.1 Statement of main results

Now we present our final bound that characterizes the performance of Algorithm 1 as Theorem 6. In the deterministic black-box optimization literature [26, 34], the quantity of interest is generally *simple regret*, $R(\Lambda)$, as defined in Equation 4. In this line of work, the simple regret scales as a function of *near-optimality dimension*, which is defined as follows:

**Definition 5.** *The near-optimality dimension of $G(\cdot)$ with respect to parameters $(\nu_1, \rho)$ is given by:*

$$d(\nu_1, \rho) = \inf \left\{ d' \in \mathbb{R}^+ : \exists \, C(\nu_1, \rho), \, s.t. \, \forall h \geq 0, \mathcal{N}_h(3\tilde{\nu}\rho_2^h) \leq C(\nu_1, \rho)\rho_2^{-d'h} \right\},$$

*where $\mathcal{N}_h(\epsilon)$ is the number of cells $(h, i)$ such that $\inf_{\boldsymbol{\alpha} \in (h,i)} G(\boldsymbol{\alpha}) \leq G(\boldsymbol{\alpha}^*) + \varepsilon$, $\rho_2 = \sqrt{\rho}$, and $\tilde{\nu} = \sqrt{\nu_1}(L + \frac{\beta\sqrt{\nu_1}}{6})$.*

The near-optimality dimension intuitively states that there are *not too many* cells which contain a point whose function values are *close* to optimal *at any tree height*. The lower the near-optimality dimension, the easier is the black-box optimization problem [14]. Theorem 6 provides a similar simple regret bound on $R(\Lambda) = G(\boldsymbol{\alpha}(\Lambda)) - G(\boldsymbol{\alpha}^*)$, where $\boldsymbol{\alpha}(\Lambda)$ is the mixture weight vector returned by the algorithm given a total SGD steps budget of $\Lambda$ and $\boldsymbol{\alpha}^*$ is the optimal mixture. The proof of Theorem 6 is in the following subsection.

**Theorem 6** (Restatement of Theorem 3 from Section 5). *Let $h'$ be the smallest number $h$ such that $\sum_{l=0}^{h} 2C(\nu_1, \rho)\lambda(l)\rho_2^{-d(\nu_1,\rho)l} > \Lambda - 2\lambda(h+1)$. Then, with probability at least $1 - \frac{1}{\Lambda^3}$, the tree in Algorithm 1 grows to a height of at least $h(\Lambda) = h' + 1$ and returns a mixture weight $\boldsymbol{\alpha}(\Lambda)$ such that*

$$R(\Lambda) \leq 4\widehat{\nu}\rho_2^{h(\Lambda)-1} \tag{36}$$

*where $\widehat{\nu} = \sqrt{\nu_1}\left(L + \frac{\beta}{8}\sqrt{\nu_1}\right)$*

Theorem 6 shows that, given a total budget of $\Lambda$ SGD steps, Mix&Match recovers a mixture $\boldsymbol{\alpha}(\Lambda)$ with test error at most $4\widehat{\nu}\rho_2^{h(\Lambda)-1}$ away from the optimal test error if we perform optimization using that mixture. The parameter $h(\Lambda)$ depends on the number of steps needed for a node expansion at different heights and crucially makes use of the fact that the starting iterate for each new node can be borrowed from the parent's last iterate. The tree search also progressively allocates more samples to deeper nodes, as we get closer to the optimum. Similar simple regret scalings have been recently shown in the context of deterministic multi-fidelity black-box optimization [34]. Note that Theorem 6 roughly corresponds to a regret scaling on the order of $\tilde{O}\left(\frac{1}{\Lambda^c}\right)$ for some constant $c$ (dependent on $d(\nu_2, \rho_2)$). Thus, when $|\mathcal{D}_{te}|$ is much smaller than the total computational budget $\Lambda$, our algorithm gives a significant improvement over training only on the validation dataset. In our experiments in Section 6 and Appendix H, we observe that our algorithm indeed outperforms the algorithm which trains only on the validation dataset for several different real-world datasets.

## E.2 Proof of main results

We begin by establishing a Lipschitz-like bound for our objective function

**Lemma 11.** *Let $L = \beta \sup_{\boldsymbol{\alpha} \in \triangle} \|\boldsymbol{w}^*(\boldsymbol{\alpha}) - \boldsymbol{w}^*(\boldsymbol{\alpha}^*)\|$. Then, taking $\hat{\boldsymbol{w}}(\boldsymbol{\alpha})$ as the model returned by SGD run as in Theorem 5, we have the following bound for any mixture $\boldsymbol{\alpha} \in \triangle$:*

$$-L\|\hat{\boldsymbol{w}}(\boldsymbol{\alpha}) - \boldsymbol{w}^*(\boldsymbol{\alpha})\|_2 \leq F^{(te)}(\hat{\boldsymbol{w}}(\boldsymbol{\alpha})) - G(\boldsymbol{\alpha})$$

$$\leq \left(L + \frac{\beta}{2}\|\hat{\boldsymbol{w}}(\boldsymbol{\alpha}) - \boldsymbol{w}^*(\boldsymbol{\alpha})\|_2\right)\|\hat{\boldsymbol{w}}(\boldsymbol{\alpha}) - \boldsymbol{w}^*(\boldsymbol{\alpha})\|_2$$

*Proof.* Observe that, for any $\alpha \in \triangle$, by $\beta$-smoothness and Cauchy-Schwarz, and recalling that $\alpha^*$ is the mixture for which $F^{(\alpha^*)} = F^{(te)}$, and $F^{(te)}(w^*(\alpha)) = G(\alpha)$,

$$F^{(\alpha^*)}(\hat{w}(\alpha)) - F^{(\alpha^*)}(w^*(\alpha))$$

$$\leq \langle \nabla F^{(\alpha^*)}(w^*(\alpha)), \hat{w}(\alpha) - w^*(\alpha) \rangle + \frac{\beta}{2} \|\hat{w}(\alpha) - w^*(\alpha)\|^2$$

$$\leq \|\nabla F^{(\alpha^*)}(w^*(\alpha)) - \underbrace{\nabla F^{(\alpha^*)}(w^*(\alpha^*))}_{=0}\| \|\hat{w}(\alpha) - w^*(\alpha)\| + \frac{\beta}{2}\|\hat{w}(\alpha) - w^*(\alpha)\|^2$$

$$\leq \left( \beta\|w^*(\alpha) - w^*(\alpha^*))\| + \frac{\beta}{2}\|\hat{w}(\alpha) - w^*(\alpha)\| \right) \|\hat{w}(\alpha) - w^*(\alpha)\|$$

$$\leq \left( \beta\|w^*(\alpha) - w^*(\alpha^*))\| + \frac{\beta}{2}\|\hat{w}(\alpha) - w^*(\alpha)\| \right) \|\hat{w}(\alpha) - w^*(\alpha)\|$$

and similarly, by $\beta$-smoothness and strong convexity, we have that

$$F^{(\alpha^*)}(\hat{w}(\alpha)) - F^{(\alpha^*)}(w^*(\alpha))$$

$$\geq \langle \nabla F^{(\alpha^*)}(w^*(\alpha)), \hat{w}(\alpha) - w^*(\alpha) \rangle + \frac{\mu}{2}\|\hat{w}(\alpha) - w^*(\alpha)\|^2$$

$$\geq -\|\nabla F^{(\alpha^*)}(w^*(\alpha)) - \underbrace{\nabla F^{(\alpha^*)}(w^*(\alpha^*))}_{=0}\| \|\hat{w}(\alpha) - w^*(\alpha)\| + \frac{\mu}{2}\|\hat{w}(\alpha) - w^*(\alpha)\|^2$$

$$\geq \left( -\beta\|w^*(\alpha) - w^*(\alpha^*))\| + \frac{\mu}{2}\|\hat{w}(\alpha) - w^*(\alpha)\| \right) \|\hat{w}(\alpha) - w^*(\alpha)\|$$

$$\geq -\beta\|w^*(\alpha) - w^*(\alpha^*))\|\|\hat{w}(\alpha) - w^*(\alpha)\|$$

Plugging in the definition of $L$ to the above bounds yields the claim. $\qquad\square$

Using Lemma 11, we can now establish the following guarantee for the Mix&Match algorithm.

**Lemma 12.** *With probability at least* $1 - \frac{1}{\Lambda^3}$, *Algorithm 1 only expands nodes in the set* $J := \cup_{h=1}^{\Lambda} J_h$, *where* $J_h$ *is defined as follows,*

$$J_h := \{nodes \ (h, i) \ such \ that \ G(\alpha_{h,i}) - 3\tilde{\nu}\rho_2^h \leq G(\alpha^*)\},$$

*where* $\tilde{\nu} = \sqrt{\nu_1}\left(L + \frac{\beta}{6}\sqrt{\nu_1}\right)$.

*Proof.* Let $A_t$ be the event that the leaf node that we decide to expand at time $t$ lies in the set $J$. Also let $\mathcal{L}_t$ be the set of leaf-nodes currently exposed at time $t$. Let $B_t = \bigcap_{(h,i)\in\mathcal{L}_t} \left\{ \|\hat{w}(\alpha_{h,i}) - w^*(\alpha_{h,i})\|_2^2 \leq \nu_1\rho^h \right\}$ denote the event that every leaf node has a model that is $\nu_1\rho^h$ close to the corresponding optimal model.

Now we have the following chain,

$$\mathbb{P}(B_t^c) = \mathbb{P}\left(\exists (h,i) \in \mathcal{L}_t : \|\hat{\boldsymbol{w}}(\boldsymbol{\alpha}_{h,i}) - \boldsymbol{w}^*(\boldsymbol{\alpha}_{h,i})\|_2^2 > \nu_1 \rho^h\right)$$

$$= \sum_{l=1}^{t/2} \mathbb{P}\left(|\mathcal{L}_t| = l, \ \exists (h,i) \in \mathcal{L}_t : \|\hat{\boldsymbol{w}}(\boldsymbol{\alpha}_{h,i}) - \boldsymbol{w}^*(\boldsymbol{\alpha}_{h,i})\|_2^2 > \nu_1 \rho^h\right)$$

$$\leq \sum_{l=1}^{t/2} \sum_{k=1}^{l} \mathbb{P}\left(|\mathcal{L}_t| = l, \ \|\hat{\boldsymbol{w}}(\boldsymbol{\alpha}_{h_k,i_k}) - \boldsymbol{w}^*(\boldsymbol{\alpha}_{h_k,i_k})\|_2^2 > \nu_1 \rho^h\right)$$

$$\leq \sum_{l=1}^{t/2} \sum_{k=1}^{l} \mathbb{P}\left(\|\hat{\boldsymbol{w}}(\boldsymbol{\alpha}_{h_k,i_k}) - \boldsymbol{w}^*(\boldsymbol{\alpha}_{h_k,i_k})\|_2^2 > \nu_1 \rho^h\right)$$

$$\overset{(a)}{\leq} \sum_{l=1}^{t/2} \sum_{k=1}^{l} \frac{1}{\Lambda^7}$$

$$\leq \frac{1}{\Lambda^5}$$

Here, $(a)$ is due to the h.p. result in Corollary 3.

Now note that, due to the structure of the algorithm, an optimal node (partition containing the optimal point) at a particular height has always been evaluated prior to any time $t$, for $t \geq 2$. Now we will show that if $B_t$ is true, then $A_t$ is also true. Let $(h,i)$ be a optimal node that is exposed at time $t$. Let $b_{h,i}^*$ be the lower confidence bound we have for that node. Let us assume that the confidence boost assigned to each node's value in $b_{h,i}$ in Algorithm 1 is $\gamma(h)$ for some function $\gamma$.

Therefore, given $B_t$ we have that, taking $L = \beta \sup_{\boldsymbol{\alpha} \in \triangle} \|\boldsymbol{w}^*(\boldsymbol{\alpha}) - \boldsymbol{w}^*(\boldsymbol{\alpha}^*)\|$

$$b_{h,i}^* = \min_{(h,i) \in \mathcal{L}_t} F^{(te)}(\hat{\boldsymbol{w}}(\boldsymbol{\alpha}_{h,i})) - \gamma(h)$$

$$\leq \min_{(h,i) \in \mathcal{L}_t} G(\boldsymbol{\alpha}_{h,i}) + \left(L + \frac{\beta}{2}\|\hat{\boldsymbol{w}}(\boldsymbol{\alpha}_{h,i}) - \boldsymbol{w}^*(\boldsymbol{\alpha}_{h,i})\|\right) \|\hat{\boldsymbol{w}}(\boldsymbol{\alpha}_{h,i}) - \boldsymbol{w}^*(\boldsymbol{\alpha}_{h,i})\|_2 - \gamma(h)$$

$$\leq \min_{(h,i) \in \mathcal{L}_t} G(\boldsymbol{\alpha}_{h,i}) + L\sqrt{\nu_1}\rho^{h/2} + \frac{\beta}{2}\nu_1\rho^h - \gamma(h)$$

Now, as long as $\gamma(h) \geq 2L\sqrt{\nu_1}\rho^{h/2} + \frac{\beta}{2}\nu_1\rho^h$, then the following inequality implies that

$$b_{h,i}^* \leq G(\boldsymbol{\alpha}^*)$$

So for a node at time $t$ to be expanded, the lower confidence value of that node $b_{h,i}$ must be lower than $G(\boldsymbol{\alpha}^*)$. Now again, given $B_t$, we have that, by choosing $\gamma(h) = 2\sqrt{\nu_1}\left(L + \frac{\beta}{2}\sqrt{\nu_1}\right)\rho^{h/2}$ (satisfying the above inequality constraint),

$$b_{h,i}^* = \min_{(h,i) \in \mathcal{L}_t} F^{(te)}(\hat{\boldsymbol{w}}(\boldsymbol{\alpha}_{h,i})) - \gamma(h)$$

$$\geq \min_{(h,i) \in \mathcal{L}_t} G(\boldsymbol{\alpha}_{h,i}) - L\|\hat{\boldsymbol{w}}(\boldsymbol{\alpha}_{h,i}) - \boldsymbol{w}^*(\boldsymbol{\alpha}_{h,i})\| - \gamma(h)$$

$$\geq \min_{(h,i) \in \mathcal{L}_t} G(\boldsymbol{\alpha}_{h,i}) - L\sqrt{\nu_1}\rho^{h/2} - \gamma(h)$$

$$\geq \min_{(h,i) \in \mathcal{L}_t} G(\boldsymbol{\alpha}_{h,i}) - \underbrace{3\sqrt{\nu_1}\left(L + \frac{\beta}{6}\sqrt{\nu_1}\right)}_{=\tilde{\nu}}\rho^{h/2}$$

Therefore, we have that $\mathbb{P}(A_t^c) \leq \mathbb{P}(B_t^c)$. Now, let $A$ be the event that over the course of the algorithm, no node outside of $J$ is ever expanded. Let $T$ be the random variable denoting the total number of evaluations given our budget. We now have the following chain.

$$\mathbb{P}(A^c) = \mathbb{P}\left(\bigcup_{T=1}^{\Lambda}\left\{\bigcup_{t=1}^{T}\{A_t^c\}\right\} \cap \{T = l\}\right)$$

$$\leq \sum_{T=1}^{\Lambda}\mathbb{P}\left(\bigcup_{t=1}^{T}\{A_t^c\}\}\right)$$

$$\leq \frac{1}{\Lambda^3}$$

as desired. □

**Lemma 13.** *Let $h'$ be the smallest number $h$ such that $\sum_{l=0}^{h}2C(\nu_1,\rho)\lambda(h)\rho_2^{-d(\nu_1,\rho)l} > \Lambda - 2\lambda(h+1)$. The tree in Algorithm 1 grows to a height of at least $h(\Lambda) = h' + 1$, with probability at least $1 - \frac{1}{\Lambda^3}$. Here, $\lambda(h)$ is as defined in Corollary 3.*

*Proof.* We have shown that only the nodes in $J = \cup_h J_h$ are expanded. Also, note that by definition $|J_h| \leq C(\nu_1,\rho)\rho_2^{-d(\nu_1,\rho)}$.

Conditioned on the event $A$ in Lemma 12, let us consider the strategy that only expands nodes in $J$, but expands the leaf among the current leaves with the least height. This strategy yields the tree with minimum height among strategies that only expand nodes in $J$. The number of SGD steps incurred by this strategy till height $h'$ is given by,

$$\sum_{l=0}^{h'}2C(\nu_1,\rho)\lambda(l)\rho_2^{-d(\nu_1,\rho)l}.$$

Since the above number is greater than to $\Lambda - 2\lambda(h'+1)$ another set of children at height $h'+1$ is expanded and then the algorithm terminates because of the check in the while loop in step 4 of Algorithm 1. Therefore, the resultant tree has a height of at least $h'+1$. □

*Proof of Theorem 6.* Given that event $A$ in Lemma 12 holds, Lemma 13 shows that at least one node at height $h'$ (say $(h',i)$) is expanded and one of that node's children say $\boldsymbol{\alpha}_{h'+1,i'}$ is returned by the algorithm. Note that $(h',i)$ is in $J_h$ and therefore $G(\boldsymbol{\alpha}_{h',i}) - 3\tilde{\nu}\rho_2^{h'} \leq G(\boldsymbol{\alpha}^*)$. Invoking the smoothness property in Corollary 2, we get that

$$G(\boldsymbol{\alpha}_{h'+1,i'}) - G(\boldsymbol{\alpha}^*) = G(\boldsymbol{\alpha}_{h'+1,i'}) - G(\boldsymbol{\alpha}_{h',i}) + G(\boldsymbol{\alpha}_{h',i}) - G(\boldsymbol{\alpha}^*)$$

$$\leq \nu_2\rho_2^{h'} + 3\tilde{\nu}\rho_2^{h'}$$

$$= 4\underbrace{\sqrt{\nu_1}\left(L + \frac{\beta}{8}\sqrt{\nu_1}\right)}_{\hat{\nu}}\rho_2^{h'}$$

□

## F    Scaling of $h(\Lambda)$ and $\lambda(h)$

In this section, we discuss how to interpret the scaling of the height function $h(\Lambda)$ from Theorem 6 and the SGD budget allocation strategy $\lambda(h)$ from Corollary 3.

Let us take $k = 0$ in Theorem 5, and assume the third term in the high probability bound is dominant: that is, for some constant $K$ large enough, taking $C = \frac{4\sqrt{2\bar{C}}K\log\Lambda^8}{\mu}$, we want to choose $\lambda(h)$ to satisfy:

$$\frac{C\log\Lambda}{\sqrt{\lambda(h) + E}} \leq \nu_1\rho_2^{h+1}. \tag{37}$$

Then, solving for $\lambda(h)$, we have that

$$\lambda(h) = \left(\frac{C\log\Lambda}{\nu_1\rho_2^{h+1}}\right)^2 - E \tag{38}$$

$$= \tilde{O}\left(\frac{1}{\rho_2^{2h}}\right) \tag{39}$$

Thus, outside of the constant scaling regime discussed in Corollary 3, we expect SGD to take an exponential (in height) number of SGD steps in order to obtain a solution that is of distance $\nu_1\rho_2^{h+1}$ from the optimal solution w.h.p. (Recall that $\rho_2 \in (0,1)$)

In light of this, we may discuss now how the depth of the seach tree, $h(\Lambda)$, scales as a function of the total SGD budget $\Lambda$. We will let

$$\lambda(h) = \begin{cases} \lambda_{const} & \text{When } h \text{ is in constant step size regime} \\ \frac{C'\log^2\Lambda}{\nu_1\rho_2^{2h}} & \text{Outside of this regime, for } C' \text{ chosen large enough} \end{cases} \tag{40}$$

We may thus solve for $h'$ from Theorem 6 as follows. Denote $h_{const}$ as the maximum height of the tree for which $\lambda(h) = \lambda_{const}$ for all $h \leq h_{const}$. Then:

$$\sum_{i=0}^{h(\Lambda)-1} 2C(\nu_1,\rho)\lambda(l)\rho_2^{-d(\nu_1,\rho)l} = 2C(\nu_1,\rho)\lambda_{const}\sum_{i=0}^{h_{const}}\rho_2^{-d(\nu_1,\rho)l}$$

$$+ 2\tilde{C}\log^2\Lambda\sum_{l=h_{const}+1}^{h(\Lambda)-1}\rho_2^{-(d(\nu_1,\rho)+2)l}$$

$$= \underbrace{2C(\nu_1,\rho)\lambda_{const}\frac{\rho_2^{-d(\nu_1,\rho)(h_{const}+1)} - 1}{\rho_2^{-d(\nu_1,\rho)} - 1}}_{T1}$$

$$+ \underbrace{2\tilde{C}\log^2\Lambda\frac{\rho_2^{-(d(\nu_1,\rho)+2)h(\Lambda)} - \rho_2^{-(d(\nu_1,\rho)+2)(h_{const}+2)}}{\rho_2^{-(d(\nu_1,\rho)+2)} - 1}}_{T2}$$

$$\overset{want}{>} \Lambda - 2\lambda(h(\Lambda))$$

Now, observe that when $h_{const} = h(\Lambda)$, then $T2 = 0$, and we need that, solving for $h_{const}$,

$$h(\Lambda) > \frac{1}{d(\nu_1,\rho)}\log_{\frac{1}{\rho_2}}\left(\frac{\rho_2^{-d} - 1}{2C(\nu_1,\rho)\lambda_{const}}(\Lambda - 2\lambda_{const}) + 1\right)$$

and thus, $h(\Lambda) = h_{const}$ scales as $O(\log_{\frac{1}{\rho_2}}\Lambda)$ w.h.p.

When $\Lambda$ is sufficiently large so that $h(\Lambda) > h_{const}$ and $h_{const}$ can be taken as a constant, we need that, for a sufficiently large constant $\hat{C}$,

$$\hat{C}\log^2\Lambda\frac{\rho_2^{-(d(\nu_1,\rho)+2)h(\Lambda)} - \rho_2^{-(d(\nu_1,\rho)+2)(h_{const}+2)}}{\rho_2^{-(d(\nu_1,\rho)+2)} - 1} \overset{want}{>} \Lambda - 2\frac{C\log^2\Lambda}{\nu_1\rho_2^{2h(\Lambda)}}$$

Solving for $h(\Lambda)$, we find that, for some large enough constant $\hat{\hat{C}}$, we must have that

$$h(\Lambda) > \frac{1}{d(\nu_1,\rho)+2}\left(\log_{\frac{1}{\rho_2}}\frac{\Lambda}{\hat{\hat{C}}\log^2\Lambda}\right)$$

and thus, in this case, $h(\Lambda)$ scales as $O\left(\log_{\frac{1}{\rho_2}}\frac{\Lambda}{\log^2\Lambda}\right)$ w.h.p.

In the context of Theorem 6, this scaling shows that the simple regret of our algorithm, $R(\Lambda)$, scales roughly as $\tilde{O}\left(\frac{1}{\Lambda^c}\right)$ for some constant $c$. Thus, in certain small validation set regimes as discussed in Remark 5, Mix&Match gives an *exponential improvement in simple regret* compared to an algorithm which trains only on the validation dataset.

---

**Algorithm 2** Mix&Match: Tree-Search over the mixtures of training datasets

---

**Input:** Real numbers $\nu_1 > 0$, $\rho_2 \in (0,1)$, and hierarchical partition $\mathcal{P}$ of $\triangle$ as specified in Corollary 2, $\nu = \sqrt{\nu_1}\left(L + \frac{\beta}{2}\sqrt{\nu_1}\right)$, total SGD budget for *entire tree search procedure* $\Lambda > 0$, initial model $\boldsymbol{w}_0 \in \mathcal{W}$, SGD height-dependent budget function $\lambda(h)$ as determined by Theorem 5 and Corollary 3.

1: Initialize search tree $\mathcal{T}_0 = \{(0,1)\}$ with initial model $\hat{\boldsymbol{w}}(\boldsymbol{\alpha}_{0,1})$ trained using SGD (from Theorem 5) on training mixture distribution $\boldsymbol{\alpha}_{0,1} \in \mathcal{P}_{0,1}$ to optimization error $2\nu_1\rho^0$.
2: Cost (Number of SGD steps used): $C = \lambda(0)$
3: **while** $C \leq \Lambda$ **do**
4:     Select the leaf $(h,j) \in \text{Leaves}(\mathcal{T}_t)$ with minimum $b_{h,j} := F^{(te)}(\hat{\boldsymbol{w}}(\boldsymbol{\alpha}_{h,j})) - 2\nu\rho_2^h$.
5:     Add to $\mathcal{T}_t$ the 2 children of $(h,j)$ (as determined by $\mathcal{P}$) by querying them using Algorithm 3.
6:     $C = C + 2\lambda(h+1)$.
7: **end while**
8: Let $h(\Lambda)$ be the height of $\mathcal{T}_t$
9: Let $i^* := \arg\min_i F^{(te)}(\hat{\boldsymbol{w}}(\boldsymbol{\alpha}_{h(\Lambda),i}))$.
10: Return $\boldsymbol{\alpha}_{h(\Lambda),i^*}$ and $\hat{\boldsymbol{w}}(\boldsymbol{\alpha}_{h(\Lambda),i^*})$.

---

---

**Algorithm 3** ExpandNode: Optimize over the current mixture and evaluate

---

**Input:** Parent node $(h,i)$ with model $\hat{\boldsymbol{w}}(\boldsymbol{\alpha}_{h,i})$, $\nu > 0$, $\rho_2 \in (0,1)$

1: // Iterate over new child node indices
2: **for** $(h',i') \in \{(h+1, 2i-1), (h+1, 2i)\}$ **do**
3:     Let $\boldsymbol{\alpha} := \boldsymbol{\alpha}_{h',i'} \in \mathcal{P}_{h',i'}$ and $\boldsymbol{w}_0 := \hat{\boldsymbol{w}}(\boldsymbol{\alpha}_{h,i})$.
4:     **for** $t = 1, ..., T := \lambda(h+1)$ (see Corollary 3) **do**
5:         $\boldsymbol{w}_t = \boldsymbol{w}_{t-1} - \eta_t \nabla f(\boldsymbol{w}_{t-1}; z_t)$ for $z_t \sim p^{(\boldsymbol{\alpha})}$.
6:     **end for**
7:     Obtain test error $F^{(te)}(\boldsymbol{w}_T)$
8:     Set node estimate: $b_{h',i'} = F^{(te)}(\boldsymbol{w}_T) - 2\nu\rho_2^{h'}$.
9:     Set final model: $\hat{\boldsymbol{w}}(\boldsymbol{\alpha}_{h',i'}) = \boldsymbol{w}_T$.
10: **end for**

---

**Remark 5** (Small validation dataset regime). *Under no assumptions on the usage of the (size $n$) validation set, only $k = O(n^2)$ queries can be made while maintaining nontrivial generalization guarantees [4, 23]. When tracking only the best model, as in [6, 16], $k$ can be roughly **exponential** in the size of the validation set. While our setting is more similar to this latter setting, a precise characterization of the sample complexity, and thus of the precise bounds on the size of the validation set, is important. Here we focus on the computational aspects, and leave the formalization of generalization guarantees in our setting to future work.*

## G   Detailed Mix&Match **Algorithm**

In this section, we present the full version of Algorithm 1 as Algorithm 2, with the constants now specified.

## H   Additional Experimental Details

The code used to create the testing infrastructure can be found at `https://github.com/matthewfaw/mixnmatch-infrastructure`. The code used to run the experiments can be found at `https://github.com/matthewfaw/mixnmatch`.

### H.1   Details about the experimental setup

All experiments were run in python:3.7.3 Docker containers (see `https://hub.docker.com/_/python`) managed by Google Kubernetes Engine running on Google Cloud Platform on n1-standard-4 instances. Hyperparameter tuning was performed primarily using the Katib framework (`https://github.com/kubeflow/katib`) using the validation error as the objective, with some additional fine-tuning performed through local tests, with performance measured through validation AUC. Since all algorithms we compare against use SGD, we use the same hyperparameters (SGD

step size, neural network architecture, etc.) across all algorithms for each dataset/plot (however, they are different across plots/datasets).

All experiments reported below are the results of an average of 10 runs, and error bars correspond to 1 standard deviation. The $x$ axis of each plot corresponds to the *intermediate* amount of SGD budget used at that measured value, and the corresponding $y$ axis is the average performance of the classifier after the indicated number of SGD iterations. That is, for each algorithm, we specify a total SGD budget (e.g., 60k for the Allstate experiment, and 200k for the MNIST experiment), and report the test AUC measured at *intermediate* intervals throughout each experiment's duration

### H.2 Details about the multiclass AUC metric

We briefly discuss the AUC metric used throughout our experiments. We evaluate each of our classification tasks using the multi-class generalization of area under the ROC curve (AUROC) proposed by [15]. This metric considers each pair of classes (i,j), and for each pair, computes an estimate for the probability that a random sample from class $j$ has lower probability of being labeled as class $i$ than a random sample from class $j$. The metric reported is the average of each of these pairwise estimates. This AUC genenralization is implemented in a recent scikit-learn release `https://github.com/scikit-learn/scikit-learn/pull/12789`. In our experiments, we use this scikit-learn implementation.

### H.3 Description of algorithms used

In the sections that follow, we will reference the following algorithms considered in our experiments. We note that the algorithms discussed in this section are a superset of those discussed in Section 6.

### H.4 Allstate Purchase Prediction Challenge – Correcting for shifted mixtures

Here, we provide more details about the experiment on the Allstate dataset [1] discussed in Section 6. Recall that in this experiment, we consider the mixture space over which Mix&Match searches to be the set of mixtures of data from Florida (FL), Connecticut (CT), and Ohio (OH). We take $\alpha^*$ to be the proportion of each state in the test set. The breakdown of the training/validation/test split for the Allstate experiment is shown in Table 2.

Here, each Mix&Match algorithm allocates a height-independent 5000 samples for each tree search node on which SGD is run. Each algorithm uses a batch size of 100 to compute stochastic gradients.

#### H.4.1 Dataset transformations performed

We note that in the dataset provided by Kaggle, the data for a single customer is spread across multiple rows of the dataset, since for each customer there some number (different for various customers) of intermediate transactions, followed by a row corresponding to the insurance plan the customer ultimately selected. We collapse the dataset so that each row corresponds to the information of a distinct customer. To do this, for each customer, we preserve the final insurance plan selected, the penultimate insurance plan selected in their history, the final and penultimate cost of the plan. Additionally, we create a column indicating the total number of days the customer spent before making their final transaction, as well as a column indicating whether or not a day elapsed between intermediate and final purchase, a column indicating whether the cost of the insurance plan changed, and a column containing the price amount the insurance plan changed between the penultimate and final purchase. For every other feature, we preserve only the value in the row corresponding to the purchase. We additionally one-hot encode the car_value feature. Additionally, we note that we predict only one part of the insurance plan (the G category, which takes 4 possible values). We keep all other parts of the insurance plan as features.

#### H.4.2 Experimental results

Figure 2 shows the results of the same experiment as discussed in Section 6. We note that there are now several variants of the Mix&Match algorithm, whose implementations are described in Table 1. We observe that, in this experiment, when running Mix&Match for the entire experiment budget, the two simplex partitioning schemes result in algorithms that all have similar performance on the

Table 1: Description of the algorithms used in the experiments

| Algorithm ID | Description |
|---|---|
| Mix&MatchCH | The Mix&Match algorithm, where the simplex is partitioned using a random coordinate halving scheme |
| Mix&MatchDP | The Mix&Match algoirhtm, where the simplex is partitioned using the Delaunay partitioning scheme |
| Mix&MatchCH+0.1Step | Runs the Mix&MatchCH algorithm for the first half of the SGD budget, and runs SGD sampling according to the mixture returned by Mix&Match for the second half of the SGD budget, using a step size 0.1 times the size used by Mix&Match |
| Mix&MatchDP+0.1Step | Runs the Mix&MatchDP algorithm for the first half of the SGD budget, and runs SGD sampling according to the mixture returned by Mix&Match for the second half of the SGD budget, using a step size 0.1 times the size used by Mix&Match |
| Genie | Runs SGD, sampling from the training set according to the test set mixture |
| Validation | Runs SGD, sampling only from the validation set according to the test set mixture |
| Uniform | Runs SGD, sampling uniformly from the training set |
| OnlyX | Runs SGD, sampling only from dataset X |
| IW-Uniform | Splits training and validation data in half. Using the first half of the data, trains a logistic regression to predict if a given sample is in the training or validation distribution. Then, using this model, assigns importance weights to each of the training samples from the second split as the odds ratio of the model. Finally, runs the Uniform algorithm, minimizing training loss *weighted by the importance weights*. |
| IW-ERM | Performs the same procedure as in IW-Uniform to compute importance weights. Instead of sampling uniformly from each training distribution, we concatenate all training datasets, and run SGD to minimize loss on this combined dataset. Note that, since the sizes of the different datasets are different, this results in a slightly different algorithm than IW-Uniform. |
| MMD | Builds a training dataset from the $K$ data sources intended to be a summary of the validation dataset by greedily maximizing the (non-private) MMD objective as described in [33]. We then run SGD to minimize loss on this summary dataset. |

Table 2: The proportions of data from each state used in training, validation, and testing for Figure 1a and Figure 2

| State | Total Size | % Train | % Validate | % Test | % Discarded |
|---|---|---|---|---|---|
| FL | 14605 | 49.34 | 0.16 | 0.5 | 50 |
| CT | 2836 | 50 | 7.5 | 42.5 | 0 |
| OH | 6664 | 2.25 | 0.75 | 2.25 | 94.75 |

test set. However, in the Mix&Match instances that commit to a single mixture after half of its budget (Mix&MatchCH+0.1Step and Mix&MatchDP+0.1Step), the partitioning strategy has a more substantial impact on performance. This is not surprising, as the Delaunay partitioning strategy forces Mix&Match to create 3 children instead of 2 as in the coordinate halving strategy. Thus, the $DP$ strategy forces more exploration, and so, for a fixed budget split, it is natural to expect the coordinate halving strategy to have better performance.

Beyond these observations, we finally note that the best Mix&Match instance achieves a performance nearly identical to the Genie algorithm which knows the correct mixture to sample a priori (as well as the OnlyCT algorithm, which has performance similar to Genie since the mixture distributions

are similar, as shown in Table 2). Additionally, this Mix&Match instance out-performs all other baselines.

Figure 2: Test One vs One AUROC for Mixture of FL, CT, and OH

## H.5 Wine Ratings

We consider the effectiveness of using Algorithm 1 to make predictions on a new region by training on data from other, different regions. For this experiment, we use another Kaggle dataset [3], in which we are provided binary labels indicating the presence of particular tasting notes of the wine, as well as a point score of the wine and the price quartile of the wine, for a number of wine-producing countries. We will consider several different experiments on this dataset.

We will consider again algorithms discussed in Table 1. Throughout these experiments, we will consider searching over the mixture space of proportions of datasets of wine from countries US, Italy, France, and Spain. Note that the Genie experiment is not run since there is no natural choice for $\alpha^*$, as we are aiming to predict on a new country.

### H.5.1 Dataset transformations performed

The dataset provided through Kaggle consists of binary features describing the country of origin of each wine, as well as tasting notes, and additionally a numerical score for the wine, and the price. We split the dataset based on country of origin (and drop the country during training), and add as an additional target variable the price quartile. We keep all other features in the dataset. In the experiment predicting wine prices, we drop the price quartile column, and in the experiment predicting wine price quartiles, we drop the price column.

### H.5.2 Predict wine prices

In this section, we consider the task of predicting wine prices in Chile and Australia by using training data from US, Italy, France, and Spain. The train/validation/test set breakdown is described in Table 3. We use each considered algorithm to train a fully connected neural network with two hidden layers and sigmoid activations, similarly as considered in [42]. We plot the test mean absolute error of each considered algorithm.

Here, each Mix&Match algorithm allocates a height-independent 500 samples for each tree search node on which SGD is run. Each algorithm uses a batch size of 25 to compute stochastic gradients.

Table 3: The proportions of data from each country used in training, validation, and testing for Figure 3

| Country | Total Size | % Train | % Validate | % Test | % Discarded |
|---|---|---|---|---|---|
| US | 54265 | 100 | 0 | 0 | 0 |
| France | 17776 | 100 | 0 | 0 | 0 |
| Italy | 16914 | 100 | 0 | 0 | 0 |
| Spain | 6573 | 100 | 0 | 0 | 0 |
| Chile | 4416 | 0 | 5 | 95 | 0 |
| Australia | 2294 | 0 | 5 | 95 | 0 |

The results of this experiment are shown in Figure 3. There are several interesting takeaways from this experiment. First is the sensitivity of Mix&Match to choice of partitioning scheme. While Mix&MatchCH outperforms the Uniform algorithm and each OnlyX algorithm, Mix&MatchDP performs poorly. Note that each node in the search tree under Delaunay partitioning can have $K$ ($= 4$ in this experiment) children, each node in the coordinate halving scheme only has two children. Thus, it seems that perhaps the Dealunay partitioning scheme is overly wasteful in its allocation of SGD budget. However, when considering the split budget Mix&Match algorithms which search for mixtures only for half of their SGD budget, and commit to a mixture for the remaining half, the performance gap between the two partitioning schemes is less noticeable.

The second interesting takeaway from this experiment is that, in contrast to the other experiments considered in this paper, in this experiment, it seems that although Mix&Match outperforms both the Uniform algorithm and OnlyX algorithm, it only matches the performance of the algorithm which trains only on the validation dataset. This highlights an important point of the applicability of the Mix&Match algorithm. Running Mix&Match makes sense only when there is insufficient validation data to train a good model.

Figure 3: Test Mean Absolute Error for Mixture of US, France, Italy, and Spain data, Predict in Chile and Australia

### H.5.3    Predict wine price quartiles

In this experiment, we consider a classification version of the regression problem considered in the last experiment. In particular, we have access to training wine data from US, Italy, Spain, and France, and wish to predict the quartile of the wine price for wines from Chile. The train/validate/test breakdown in given in Table 4. We use each algorithm to train a fully connected neural network with 3 hidden layers and ReLU activations, and evaluate based on the One vs One AUROC metric described in [15]. The experimental results are shown in Figure 4.

Here, each Mix&Match algorithm allocates a height-independent 1000 samples for each tree search node on which SGD is run. Each algorithm uses a batch size of 25 to compute stochastic gradients.

Table 4: The proportions of data from each country used in training, validation, and testing for Figure 4

| Country | Total Size | % Train | % Validate | % Test | % Discarded |
|---------|-----------|---------|-----------|--------|-------------|
| US | 54265 | 15 | 0 | 0 | 85 |
| France | 17776 | 100 | 0 | 0 | 0 |
| Italy | 16914 | 100 | 0 | 0 | 0 |
| Spain | 6573 | 100 | 0 | 0 | 0 |
| Chile | 4416 | 0 | 5 | 95 | 0 |

We observe that each instance of Mix&Match outperforms both Uniform and Validation (which has quite poor performance in this experiment), and has competitive performance with the best OnlyX algorithm. Additionally, Mix&Match outperforms the importance-weighted baselines, and either outperform or match the performance of the MMD algorithm.

Figure 4: Test One vs One AUROC for Mixture of US, France, Italy, and Spain data, Predict in Chile

## H.6 Colored MNIST

Here, we consider a setup similar to the Colored MNIST experiment from [2]. In particular, we consider the following classification problem: we wish to train a classifier for MNIST handwritten digits to detect whether the digit is smaller than $5$ or larger than $5$. We flip the label of each image independently with probability $0.25$. We then color each image one of two colors, determined by the image's label. Finally, we flip the color of each image with probability $e \in (0, 1)$, where $e$ is a parameter that is different for each environment.

Under this setup, a classifier which ignores the color of the image should have accuracy roughly $0.75$ (since we flip the label of the image with probability $0.25$). Depending on the choice of parameter $e$, classifiers can potentially achieve a higher accuracy by exploiting the color correlation from their own environment. However, by exploiting these spurious correlations, the classifier performance will suffer in other environments, where the color correlation can be potentially flipped.

In this setting, we cannot hope that, in general, Mix&Match will find an invariant predictor, as, by construction, *any* model output by Mix&Match will be a model trained on *some* mixture distribution over the training sets. If every training environment provided to Mix&Match exploits the same color correlation, then so will the model output by Mix&Match.

Hence, in this setup, we will consider training environments that are *sufficiently diverse* – that is, there are training environments which are both negatively *and* positively correlated with color. However, in each of the following experiments, there are *more* environments which have positive color correlation than negative. Hence, in this setting, mixture search guided by validation loss results in increased performance over simply sampling uniformly over training datasets.

### H.6.1 Dataset transformations performed

The MNIST dataset was obtained using the built-in MNIST dataset wrapper. For each image, we undersample the pixels by $4x$ for computational convenience. We then construct the label for each image by thresholding the original $0 - 9$ label by checking if the label is smaller than or larger than $5$, and discard the original label. We then flip each label independently with probability $0.25$. We then each assign each image a color ($0$ or $1$) according to the image's label. Depending on the environment, we flip this color with probability $e$ for environment $e$. We emulate coloring the image by, for each image, doubling the number of features by creating duplicate columns of each original feature. Images colored $1$ have *all* entries in the duplicated columns set to $0$. For images colored $0$, all duplicated columns are kept unchanged.

### H.6.2 Model setup

As in [2], we use a neural network with $1$ hidden layer with $390$ hidden neurons, and ReLU activations. The loss function used is the binary cross-entropy loss, with a logit applied to the output of the network.

Table 5: The proportions of data from each color environment used in training, validation, and testing for Figure 5

| Environment | Total Size | % Train | % Validate | % Test | % Discarded |
|---|---|---|---|---|---|
| 0.1 | 20000 | 100 | 0 | 0 | 0 |
| 0.2 | 20000 | 100 | 0 | 0 | 0 |
| 0.7 | 20000 | 100 | 0 | 0 | 0 |
| 0.9 | 10000 | 0 | 10 | 90 | 0 |

(a) Validation

(b) Test

Figure 5: AUROC for predicting Colored MNIST digit $< 5$ where color in validation and test environment flipped with probability $0.9$

### H.6.3    Both validation and test color strongly negatively correlated with label

In this experiment, we have 3 training environments, where the probability of flipping the color is $\{0.1, 0.2, 0.7\}$. In the validation and testing environments, the probability of flipping the color is $0.9$. The data configurations are described in Table 5.

Each Mix&Match algorithm is implemented to have a height-independent SGD budget of $5000$ iterations/samples. Each gradient step is computed using a minibatch of $100$ samples. The results of these experiments are reported in Figure 5.

Since Mix&Match does not have access to a training environment with $e = 0.9$, the output classifier cannot exploit the color correlation as effectively as the Validation classifier. However, we observe that Mix&Match is able to recover similar performance to the classifier trained only in environment $e = 0.7$, and outperform the other baselines.

### H.6.4    Validation color strongly negatively correlated with label, test color uncorrelated

As in the previous experiment, we have the same 3 training environments, where the probability of flipping the color is $\{0.1, 0.2, 0.7\}$. In the validation and testing environments, however, are now *different*, with the probability of flipping the color is $0.9$ in validation, but $0.5$ in test. The data configurations are described in Table 6.

Table 6: The proportions of data from each color environment used in training, validation, and testing for Figure 6

| Environment | Total Size | % Train | % Validate | % Test | % Discarded |
|---|---|---|---|---|---|
| 0.1 | 20000 | 100 | 0 | 0 | 0 |
| 0.2 | 20000 | 100 | 0 | 0 | 0 |
| 0.5 | 9000 | 0 | 0 | 100 | 0 |
| 0.7 | 20000 | 100 | 0 | 0 | 0 |
| 0.9 | 1000 | 0 | 100 | 0 | 0 |

(a) Validation

(b) Test

Figure 6: AUROC for predicting Colored MNIST digit $< 5$ where color in validation is flipped with probability $0.9$, but color in test environment is flipped with probability $0.5$

Each Mix&Match algorithm is implemented to have a height-independent SGD budget of $5000$ iterations/samples. Each gradient step is computed using a minibatch of $100$ samples. The results of these experiments are reported in Figure 6.

Here, we observe that, by training on several different environments instead of only an environment with a single strong correlation, Mix&Match is able to perform well in both validation and test environments, where the probabilities of flipping color are quite different. The other classifiers, by contrast, exhibit more drastic performance differences.

### H.6.5 Both validation and test colors weakly negatively correlated with label

Table 7: The proportions of data from each color environment used in training, validation, and testing for Figure 1b and Figure 7

| Environment | Total Size | % Train | % Validate | % Test | % Discarded |
|---|---|---|---|---|---|
| 0.1 | 20000 | 100 | 0 | 0 | 0 |
| 0.2 | 20000 | 100 | 0 | 0 | 0 |
| 0.6 | 10000 | 0 | 10 | 90 | 0 |
| 0.7 | 20000 | 100 | 0 | 0 | 0 |

(a) Validation

(b) Test

Figure 7: AUROC for predicting Colored MNIST digit $< 5$ where color in validation and test environment flipped with probability $0.6$

Again, we consider the same 3 training environments, where the probability of flipping the color is $\{0.1, 0.2, 0.7\}$. In the validation and testing environments here are the same, with probability of flipping color $0.6$. The data configurations are described in Table 7.

Each Mix&Match algorithm is implemented to have a height-independent SGD budget of $5000$ iterations/samples. Each gradient step is computed using a minibatch of $100$ samples. The results of these experiments are reported in Figure 7.

In this setting, the color correlation with the label is weaker than in the prior setup. Thus, we observe that, by searching over mixture distributions, Mix&Match is able to outperform all other baselines, and obtain performance competitive with the classifier trained only on the environment with the same color correlation as the validation/test sets.

## Footnotes

[1]Note that the step size schedule in Lemma 1 takes a larger value of $E$ than used in Theorem 2 from [27]. However, their results continue to hold for this choice of $E$, as noted in page 13 of their proof of this theorem.

[2]Note that here, we are considering *projected* SGD, while the analysis in [27] is done without projection. However, the proof of Theorem 2 trivially continues to hold under projection, as a result of the inequality $\|\Pi_{\mathcal{X}}(\tilde{\boldsymbol{w}}_{t+1}) - \boldsymbol{w}^*\|^2 \leq \|\tilde{\boldsymbol{w}}_{t+1} - \boldsymbol{w}^*\|^2$ (see Lemma 3.1 in [8]), for example.

[3]See Remark 3 for a discussion on our reasoning for using a global diameter bound here.