[Reviews · NeurIPS 2020]

Review 1

Summary and Contributions: In this paper, the authors address the problem of identifying the optimal mixture of distributions over multiple training sets, that has maximal performance on a validation set. The difficulty are that the distributions of data might differ between each set, which is compounded by the presence of unobserved variables. The authors introduce a new algorithm,Mix&Match, based on hierarchical bandits and stochastic gradient descent, that solves this problem, and prove an upper bound on the simple regret that it incurs. Mix&Match is then empirically evaluated and compared to existing methods.

Strengths: - The problem of identifying the best mixture in the presence of unobserved variables is new (as far as I know) and is interesting. - The upper bound on the regret of the SGD (Theorem 2/5) is an interesting contribution - The solution proposed by the authors is insightful - Overall the paper is well written and pleasant to read

Weaknesses: - The hypotheses required by Mix & Match are very strong. Proposition 1 assume thats there exists a mixture of the training distribution that perfectly matches the validation distribution. This seems very limiting to me, in particular as it requires the different training set to be correctly split -- which is impossible to verify in practice, since there are unobserved variables. - Similarly, the clustering argument of the authors (line 184 - 186) is unclear to me due to the unobserved variables. - In the experiment section, Mix& Match appears to frequently outperform the Genie algorithm, depending on the SGD budget. But Genie is an "oracle" method, that has access to the true mixture, so how is it possible ? - Figure 1a is very difficult to read

Correctness: All the claims appear to be correct, up to many typos that makes the statements difficult to follow. For instance, Theorem 2 appears to prove a lower bound on the regret instead of an upper bound, and the definition of \beta-smooth in the appendix is false.

Clarity: The paper is well written, but there are multiple errors/ typos in the statements.

Relation to Prior Work: The relationship between this work and the literature are clearly discussed.

Reproducibility: Yes

Additional Feedback: [After Rebuttal] Thank you for the clarifications. The authors adressed some of my concerns, however I think that the argument of "simply adding more distributions and re-run" does not fully address my main concern as this approach drastically increases the model space. Therefore I choose to not increase my score.


Review 2

Summary and Contributions: This paper considers the problem of dealing with the covariate shift in machine learning. The setup is that we are given a set of training data distributions and a validation data distribution. The key assumption is that the validation dataset is a covariate-shifted version that takes the form of a particular mixture of train data distribution. This is defined in a way that allows unobserved features. This setting is more general and flexible than previously proposed ones because those can fail in this setup. The authors propose an algorithm that combines SGD with tree search, and prove its convergence.

Strengths: The problem of covariate shift (or distributional robustness) is important, and the authors provide one method with the computational complexity in mind. I find it relevant and important, and the solution seems novel. The new h.p. bound on SGD is interesting technically.

Weaknesses: The particular formulation of forming a tree search may not be the best thing to do, but it seems quite competitive among available algorithms as of now.

Correctness: Yes

Clarity: Yes. I found numerous discussions to support the claim, defend the weakness, and distill the contribution.

Relation to Prior Work: yes.

Reproducibility: Yes

Additional Feedback: (after the rebuttal) I've read the rebuttal and keep the same score. The authors responded that "The emphasis in Theorem 2 is on the scaling with respect to d0 and G∗", but it's not quite clear in the current form with G undefined, etc. I suggest that the authors find ways to show distill the dependence on d_0 and G^*, and connect it to the exponentially decaying term in d_0^2 to make the point clear. ==== The paper is well-written. I found the problem setting quite interesting. The h.p. bound on the SGD convergence where the dependence on the stochastic gradient norm only at the optimal solution is interesting (though I am not expert in these algorithms).   Q: Could there be a way to form some sort of alternating minimization between w and alpha? If theoretical statements on this are hard, at least can we see how it works empirically?   The form of Theorem 2: I would like to see a simpler and more modular form of the statement. First, I would not directly use \Lambda here so that the contribution on SGD itself is decoupled from the overall computational budget which is needed only for the covariate shift problem here. Could the authors restate it so we can clearly see the dependence on \kappa, \delta (the failure rate of the probabilistic statement "w.p. >= 1-\dt"), t, ||w_0 - w^*||, and \mathcal{G}_* ?   comments • Abstract: To me, it was hard to identify the 'interesting part' from the abstract. I would mention that the presence of unobserved feature is something that was not dealt with before, but more general and cannot be handle from existing work.  • L515: "Definition Definition 1" • L521: isn't the inequality the other direction..? • Isn't the strong convexity assumption too strong, given that we assume those datasets are all from a different distribution? In the linear model with squared loss, the data must span the whole space. We also must know \mu to tune the learning rate, and the convergence is slow when \mu is small. When dealing with one dataset, we can preprocess it well, but for multiple datasets, I am not sure if we can do it. Please comment on this. • Theorem 2: It seems to me that the dependence on large \kappa means faster convergence at first sight, because the convergence scales like poly(1/E)? Please specify the dependence on \kappa. • Smoothness of G(): I was expecting to see something that bounds || \nabla G(\alpha) - \nabla(\alpha') || to check the smoothness. But I can only find something about |G(\alpha) - G(\alpha')|, which is more like Lipschitzness. Could the authors explain more on this? • L260: "that that" • L266: I am not sure if the explanation here suffices to explain why the error can decay exponentially in h+1. Explain a bit more please. • Theorem 2: Is G(.,.) explained somewhere? Note that G() was used before, so it seems like a notation clash. • Algorithm 1: Was T_t defined before? Also, I don't know where this variable t came from. Is this some sort of iteration number? • L251: doesn't this comment conflict with the one in L289 (and the choice of \lambda(h) = \lambda in the appendix) • I was surprised to see the naïve implementation of mix&match is not shown. How does it work and why was it excluded? If there is a reason why it did not work well, what's the authors' guess? I think this is something worth being discussed in the main text.


Review 3

Summary and Contributions: Given access to K datasets with distributions $(p_i)_{i=1}^K$, the authors consider the problem of finding the best mixture of these distributions, such that a learning algorithm trained on the mixture distribution performs well on a target distribution (p^{(te)}). _____________________________________________________________ AFTER FEEDBACK: I thank the authors for their clarifications regarding the requirement on the validation set size. Based on the authors' feedback and the comments of other reviewers I am happy to upgrade my score to 7. _____________________________________________________________ Under the assumption that the target distribution can be written as a mixture of the K given distributions, the authors frame the problem as a noisy black-box optimization problem over the (K-1) dimensional simplex. For every value of a mixture vector $\alpha$ in the simplex, the proposed approach runs SGD for a certain number of steps. Under strong convexity and smoothness assumptions on the loss function, the authors show that the optimization objective function is Lipschitz continuous. Next, the authors derive a bound on the suboptimality of a solution after $t$ steps of running SGD. These two results together allow the authors to use the ideas from Lipschitz black-box function optimization algorithms, such as HOO of Bubeck et al. 2011, to design an algorithm to find the best mixture vector given a total computational budget of $\Lambda$ SGD steps. The authors derive a bound on the Simple regret of the proposed algorithm, and also demonstrate its good performance over various baselines in several empirical tasks.

Strengths: The key technical contribution of this work, in my opinion, is to set up the task of estimating the best mixture vector $\alpha^*$ as that of optimizing a Lipschitz continuous black-box function which can be accessed via 'noisy' observations, where the amount of 'noise' can be controlled (i.e., by varying the number of SGD steps). This involves two results: 1. The first result (Theorem~1 in Section~5 and Theorem~4 and Corollary~2 in Appendix~C) demonstrate the Lipschitz continuity of the optimal weights (w^*(\alpha)) and the optimization objective function $G(\alpha)$. 2. An improved high probability bound on the potential function $\|w_t - w^*\|_2^2$ after $t$ steps of SGD. I feel that the proofs of these two results are the main technical contributions of this work. Once these results have been established the design of the algorithm (Mix&Match) and its analysis follow along somewhat standard lines. Besides the theoretical contributions, the authors also demonstrate the good performance of the proposed algorithm empirically on two datasets.

Weaknesses: The setup of the problem seems a bit contrived to me. The authors assume that the size of the validation set is smaller than the computational budget and hence it is infeasible to perform SGD directly on the validation set. At the same time, they also assume that the validation set is large enough that the $F^{(te)}(.)$ value can be obtained accurately *uniformly for all w* trained after at least one SGD step. Since most non-trivial ML models have very large parameter size (i.e., w lies in high dimensions) and the fact that the SGD guarantees are dimension-free, it seems to me that the number of samples required to ensure the accurate oracle access to $F^{(te)}$ might be larger than those required to train a model using SGD. It would be very helpful if the authors justify the above two seemingly contradictory assumptions on the size of the validation dataset, in case I have misinterpreted them.

Correctness: I checked the proof of the continuity results (Theorem~4 and Corollary~2 in Appendix~C) and they look correct to me. I did not have the time to go through the other proofs in details. The methodology used in the experiments look sound to me. The authors describe the details of all the baseline methods used, and report their performance along with the appropriate error bars.

Clarity: Yes.

Relation to Prior Work: Yes.

Reproducibility: Yes

Additional Feedback: Minor Comments: Is there a typo in the statement of Theorem~2? Shouldn't it be $\|w_{t+1} - w^*\|_2^2$ instead of $\|w_{t+1}-w_0\|_2^2$? In Assumption~1, is F^{(\alpha)} assumed to be a strongly-convex function for all possible choices of (p_i)_{i=1}^K? It would be helpful if the authors include some examples to demonstrate the gain in generality achieved by making this assumption, in comparison to assuming that the individual loss functions $f(\cdot, z)$ is convex.


Review 4

Summary and Contributions: The paper proposes a new algorithm for one particular setup of transfer learning: the training data is partitioned into N distributions and there is a small validation set of the target distribution; the goal is to do well on the validation set assuming its distribution is well modeled as an unknown mixture of the training distributions. The method proposed in this paper does provably almost as well as any mixture (it presents a regret bound). The method works by using a hierarchical partition of the simplex of distributions over the datasets and using warm-started SGD on mixture distributions to evaluate candidate distributions in a tree search algorithm.

Strengths: At first value the result is surprisingly powerful; there are many transfer learning settings which are approximately solvable as mixtures of known distributions and it's rare to find mathematical guarantees of this strength. The algorithm itself is fairly simple; it repeatedly picks a node in the search tree to explore and explores it by doing a small number of SGD updates and computing the validation loss of those models. In the experiments the proposed algorithm performs in practice in line with the theoretical expectations, coming close to the oracle performance when the distribution is known and outperforming reasonable baselines when the distribution isn't known.

Weaknesses: The way the paper is written somewhat hides the exponential complexity of the algorithm as a function of the number of training distributions. This is hidden AFAICT in the near-optimality dimension of the loss function over mixtures. The paper also glosses over the procedure for bisecting the simplex, which is a key part of the algorithm since the bisection has to preserve the metric very well for the results to be practically useful (or there will be a lot of backtracking during the search). This lack of clarity about the algorithm would make it quite hard to reproduce the results without going over the code. The code was provided, however. The experimental results are somewhat suspicious. Specifically, in figure 1a (though the "1" is not present in the text) we see in much of the budget space the non-optimal strategy "OnlyCT" which trains on data from only one domain outperforming the strategy "Genie" which is the oracle strategy with knowledge of the true data distribution (and similarly you can see one mixmatch variant outperforming genie). The text makes no attempt to explain this counterintuitive result; my best guess is that the baselines were not properly tuned. Not tuning the baselines in figure 1a makes me not trust the experimental results in figure 1a and also in figure 1b since it's not clear whether those trend lines would revert if proper experimental practices were adopted.

Correctness: As far as I can tell the paper is correct, though I did not thoroughly verify the proofs as they run through many pages of the supplemental material. The overall gist of the results is believable, though. I wish the paper was a little more clear about the limits of the applicability of this method. Things like what are the conditions on the data distribution for the performance with a reasonable budget to be good? Why would a practitioner not always divide a training dataset of N examples into N independent domains for maximum flexibility?

Clarity: The paper is generally well written.

Relation to Prior Work: No issues here.

Reproducibility: No

Additional Feedback:

[Author Response · NeurIPS 2020]

**Reviewer 1**: *On the assumptions in Proposition 1.* Note that, although we require such a mixture to exist, we do *not* require this mixture to be known (hence the mixture search portion of our algorithm). In practice, one could begin with a set of distributions, run Mix&Match to find the mixture distribution with smallest validation loss, and if the model does not have high enough accuracy, simply add more distributions and re-run Mix&Match. Note also that, to our knowledge, there are no other known techniques that can provably correct for distribution shift when the shift is due in part to changes in latent variables (see Remark 1). Our framework also permits shifts in $p(y|x)$, which cannot be tolerated under the common covariate shift assumption. Therefore, while such an assumption need not always be true, it allows us to prove results in a nontrivial setting, and additionally seems to be quite an effective heuristic in practice, as we demonstrate in several experiments in Section 6 and Appendix H.

*Regarding comments on experiments*: Refer to the response for Reviewer 4.

*Lower bound on regret*: Assuming you mean Theorem 3 here – the theorem is correct as stated. Recall that we are solving a minimization problem.

**Reviewer 2**: *On typo in $\beta$-smooth definition*: Yes, this was a typo. We however use the correct defn. in all of our proofs.

*Strong convexity assumption*: While ideally we could relax this assumption, it allows us to prove theorems for a variety of distribution shifts that existing techniques cannot provably correct. Additionally, this assumption does not appear to limit the practical applicability. Indeed, our algorithm performs well in practice when training neural networks on a variety of problems, as we demonstrate in our experiments.

*Theorem 2 scaling with $\kappa$.* Larger $\kappa$ will not imply a faster convergence rate, as there is a $\kappa$ dependence in the third term in $\tilde{C}$. The emphasis in Theorem 2 is on the scaling with respect to $d_0$ and $\mathcal{G}_*$, since Mix&Match aims to reuse models to get an exponentially decaying term in $d_0^2$.

**Smoothness of** $G$. We mean Lipschitz continuity, as we want close-by models to imply the solution values are close.

$G(.,.)$ **in Theorem 2**. Yes, this is a clash in notation. The use of this term is meant to follow the notation in Bottou et. al., 2018. It is defined in the formal statement of Theorem 2 (Theorem 5 in the appendix).

*L251+L266 comment*. The key point is that, by reusing models from the parent node, by Corollary 1, the $d_0^2$ term decays exponentially with height. Thus, as long as this term is large relative to the noise of the stochastic gradient, it is sufficient to take a number of steps to reach the error guarantee required by our algorithm. Beyond this point, however, the SGD budget for a node must scale with tree height.

**Reviewer 3**: *Validation set size*: The constraint that the validation loss can only be queried after using $\geq 1$ SGD step simply ensures that, in our model, an algorithm which queries the validation loss infinitely many times without using *any* computational budget is *disallowed*. We do not require that the validation loss can be obtained accurately uniformly over all models – we only need this guarantee for the models that our guarantees require, which is much fewer. Additionally, as the search tree grows deeper, models along a given path in the tree become increasingly similar, and have similar loss (Corollary 1). Thus, we can leverage results such as the recent work "Model Similarity Mitigates Test Set Overuse," Mania et. al. 2019.

$\boldsymbol{w}^*$ *vs* $\boldsymbol{w}_0$ *in Theorem 2*. Yes, this is a typo and should be $\|\boldsymbol{w}_{t+1} - \boldsymbol{w}^*(\boldsymbol{\alpha})\|^2$.

*Strong convexity*. We assume also that $f(.,z)$ is convex. We use strong convexity of the averaged distribution to obtain SGD concentration results on the $\ell_2$ distance between the final iterate and optimal solution (Theorem 2), and then also to argue that close mixtures imply close models (Theorem 1).

**Reviewer 4**: *Environment scaling+partitioning*: For more insights in scaling with respect to number of environments $K$, please refer to Corollary 2 in the appendix. This also provides a reference for the simplex bisection strategy. We will add more details addressing these issues in the main body of the paper.

*Experiments*: In the Allstate experiment (Figure 1a), the mixture is mostly ($\sim 93\%$) CT data (see Table 2 in the appendix). Thus, it seems reasonable that OnlyCT outperforms the Genie during earlier iterations when features from minority classes are less likely to be useful. For each plot, we run *all algorithms with the same hyperparameters* (SGD step size, neural network architecture, etc.). Since all algorithms we compare against *use SGD with identical parameters running simply on different mixture distributions*, we view this as a fair comparison point. Additionally, after $60k$ SGD iterations, `Genie` *does* outperform all other algorithms. We chose to include the intermediate measurement points before $60k$ iterations to increase transparency of the performance of each algorithm over time.

The hyperparameters are listed in the shell scripts in the `experiment_running` folder. The Allstate parameter settings are in `allstate_aimk_alt_newfeats_alt2.sh`. For example, in this file, the variable `NU` configures the step size for every experiment, and `BATCH` configures the SGD batch size. Line $42$ of this file runs the `Genie` experiment. This script sets up the necessary parameters to run the python script for the experiments, `run_single_experiment.py`.

[Meta-Review · NeurIPS 2020]

The reviewers generally liked this paper and also provided a number of suggestions for improvement. Please take these recommendations seriously when revising the paper. In particular, I agree with Reviewer 4 that the informal theorem statements in the main body obscure many details. Theorem 2, in particular, seems to be simultaneously too formal (do we need all these exact numeric constants?), while also obscuring important details. Overall, the ideas are interesting but I found the paper somehow a bit messy to read. Impact will be maximised by cleaning this up a for the final version.